# Impact of gravity waves on the motion and distribution of atmospheric ice particles

Aurélien Podglajen[1], Riwal Plougonven[1], Albert Hertzog[2], and Eric Jensen[3]

[1]Laboratoire de Météorologie Dynamique/IPSL, École Polytechnique, Paris-Saclay University, Palaiseau, France
[2]Laboratoire de Météorologie Dynamique/IPSL, UPMC University Paris 06, CNRS, Palaiseau, France
[3]NASA Ames Research Center, Moffett Field, California

*Correspondence to:* Aurélien Podglajen (aurelien.podglajen@lmd.polytechnique.fr)

**Abstract.** Gravity waves are an ubiquitous feature of the atmosphere and influence clouds in multiple ways. Regarding cirrus clouds, many studies have emphasized the impact of wave-induced temperature fluctuations on the nucleation of ice crystals. This paper investigates the impact of the waves on the motion and distribution of ice particles, using the idealized 2-D framework of a monochromatic gravity wave. Contrary to previous studies, a special attention is given to the impact of the wind field induced by the wave.

Assuming no feedback of the ice on the water vapor content, theoretical and numerical analyses both show the existence of a *wave-driven localization* of ice crystals, where some ice particles remain confined in a specific phase of the wave. The precise location where the confinement occurs depends on the background relative humidity, but it is always characterized by a relative humidity near saturation and a *positive vertical wind anomaly*. Hence, the wave has an impact on the mean motion of the crystals and may reduce dehydration in cirrus by slowing down the sedimentation of the ice particles. The results also provide a new insight into the relation between relative humidity and ice crystals presence.

The wave-driven localization is consistent with temperature-cirrus relationships recently observed in the tropical tropopause layer (TTL) over the Pacific during the Airborne Tropical Tropopause EXperiment (ATTREX). It is argued that this effect may explain such observations. Finally, the impact of the described interaction on TTL cirrus dehydration efficiency is quantified using ATTREX observations of clouds and temperature lapse rate.

## 1 Introduction

Atmospheric gravity waves have long been reckoned to interact with cirrus clouds. They generate temperature fluctuations, with negative anomalies favoring ice particle formation (e.g. Potter and Holton, 1995). High frequency gravity waves influence the cooling rates undergone by air parcels, which has an overwhelming impact on the properties of newly nucleated clouds (Jensen et al., 2010; Spichtinger and Krämer, 2013; Dinh et al., 2016; Jensen et al., 2016). So far, most studies investigating the impact of waves on ice clouds have been focusing on temperature anomalies. However, gravity waves also have a wind signature, and might move around sedimenting ice particles in a different way than they do air parcels. This in turn could modulate the life time of the crystals, but also the efficiency of dehydration by cirrus. Indeed, dehydration is achieved when ice crystals grow and travel a significant distance on the vertical before they start to sublimate (Dinh et al., 2014; Podglajen et al.,

2016b). Depending whether the vertical winds induced by the waves are opposing or accelerating the sedimenting motion of the particles, they might diminish or enhance the efficiency of water redistribution by sedimentation.

The goal of this paper is hence to examine the influence of internal gravity waves on ice crystal transport. Motivated by recent observations of wave-temperature relations in the Tropical Tropopause Layer (TTL) by Kim et al. (2016), we focus on the temperature range of TTL cirrus clouds (around 190 K). However, the theory presented is general and might apply to a number of aerosol particles present in different regions of the atmosphere affected by gravity waves, among which particles in Polar Stratospheric Clouds, Noctilucent Clouds in the mesosphere, or even stratospheric aerosols. Incidentally, the study will also bring a deeper insight into the relation between relative humidity and ice crystals' presence.

The article is organized as follows. In Sect. 2, a simplified setting is used to investigate the wave impact on ice crystal transport analytically. Then, in Sect. 3, the relevance of the analytical results is tested using numerical simulations and the impact of the described effect is investigated in observations of TTL cirrus during ATTREX. Implications are discussed in Sect. 4. Finally, Section 5 provides the conclusions.

## 2 Theory

To leading order, propagating waves are not expected to affect transport, as they reversibly slosh fluid parcels to and fro. Yet, the second-order Stokes drift (e.g. Andrews et al., 1987) can lead to irreversible transport. For internal gravity waves or for equatorial Kelvin waves in the Boussinesq approximation (neglecting the decrease of density with altitude), however, that Stokes drift term cancels out. There is no mean transport of a purely Lagrangian tracer by a monochromatic internal gravity wave. However, ice crystals (or aerosols) are *not* purely Lagrangian tracers. In the vertical, they fall relative to the surrounding air. Since the wave phase generally propagates downward (if the energy is to propagate up above wave sources), the falling particles fall in the direction of wave propagation. If the particles are falling at the same speed as the wave propagates downward, they will remain in the same wave phase, and hence see a constant wind anomaly: thus, there is potentially a systematic effect of the wave presence on the mean motion of the particles.

In the following, we use a simple 2D framework with the wind and temperature structure of a monochromatic wave within an unsheared background $\left(\frac{d\bar{u}}{dz} = 0\right)$ to examine the potential effects of the wave on ice crystals transport. In order to simplify the notations and without loss of generality, we furthermore assume that there is no background wind. In this idealized set-up, there is an exact analytical solution for the wave disturbance, which renders analytical progress possible. Furthermore, despite this idealization, the assumptions are not completely unrealistic: although shear can be large in the atmosphere, a significant part of it can be attributed to the waves themselves rather than to the background flow (e.g. Podglajen et al., 2017); quasi monochromatic waves have long been observed in the atmosphere, such as Kelvin waves in the TTL (e.g. Wallace and Kousky, 1968; Boehm and Verlinde, 2000), gravity waves in the mesosphere (Rapp et al., 2002), or mountain waves in the upper troposphere and stratosphere. We would also like to emphasize that the qualitative results obtained through the investigation of this idealized case are based on robust properties of the wave-ice crystal system and probably apply to more complex flows.

## 2.1 Constant size particle

First, consider the case of a constant size particle, which is assumed to sediment vertically with a downward speed $v_{\text{sed}} > 0$. The evolution of the particle's position $X(t)$, $Z(t)$ is then given by:

$$\frac{\mathrm{d}X}{\mathrm{d}t} = U\cos(kX + mZ - \omega t + \phi) \tag{1}$$

$$\frac{\mathrm{d}Z}{\mathrm{d}t} = W\cos(kX + mZ - \omega t + \phi + \delta\phi) - v_{\text{sed}} \tag{2}$$

where $U$ and $W$ are the amplitude of the wave in horizontal and vertical wind respectively, $\omega$ is the frequency, $k$ the horizontal wavenumber, $m$ the vertical wavenumber and $\phi$ the wave phase. Note that we consider a gravity wave in the midfrequency range ($f \ll \omega \ll N$ with $f$ the local Coriolis frequency and $N$ the Brunt-Väisälä frequency), so that the polarization relations (e.g. Fritts and Alexander, 2003) impose that the wave horizontal wind perturbation is aligned with the horizontal wavenumber. The 2D-plane $x - z$ is chosen along the direction of propagation of the wave (and is not necessarily zonal). Near the equator ($f \to 0$), these formulas also describe equatorial Kelvin waves and $x - z$ is then a vertical-zonal plane. For both types of waves, the polarization relations also give:

$$W = -\frac{k}{m}U \text{ and } \delta\phi = 0. \tag{3}$$

Then, there is an analytical formula for the vertical trajectory $Z(t)$ of the particle with initial position $X_0, Z_0$:

$$Z(t) = \frac{W}{\omega + mv_{\text{sed}}}\left\{\sin((\omega + mv_{\text{sed}})t)\cos(\phi_0) + \sin(\phi_0)[1 - \cos((\omega + mv_{\text{sed}})t)]\right\} - v_{\text{sed}}t + Z_0 \tag{4}$$

with $\phi_0 = kX_0 + mZ_0 + \phi$. This analytical solution highlights the difference with a no-wave case, which would simply give $Z_{\text{nowave}}(t) = -v_{\text{sed}}t + Z_0$, but also with a Lagrangian air parcel, whose vertical position is given by:

$$Z_{\text{parcel}}(t) = \frac{W}{\omega}\left\{\sin(\omega t)\cos(\phi_0) + [1 - \cos(\omega t)]\sin(\phi_0)\right\} + Z_0. \tag{5}$$

As gravity waves in the upper troposphere mostly propagate from lower levels, their vertical group velocity is positive which implies that their vertical phase speed $c_{\phi_z} = \frac{\omega}{m}$ is negative (Fritts and Alexander, 2003). In the following, we take the convention $\omega > 0$ so that a negative vertical phase speed is associated with $m < 0$. Depending on the ratio between $c_{\phi_z}$ and $v_{\text{sed}}$, different cases may arise. If $|c_{\phi_z}| \gg v_{\text{sed}}$, the particles follow the air parcels and oscillate vertically. The particles are close to perfectly Lagrangian, and there is not net effect of the wave. If $|c_{\phi_z}| \ll v_{\text{sed}}$, the particles travel in a stationary wave field, i.e. through positive and negative wave phases, whose contributions cancel out in a long enough temporal average. The interesting interaction appears when $c_{\phi_z}$ and $v_{\text{sed}}$ are of the same order: then, as $\omega + mv_{\text{sed}} < \omega$ and $\omega + mv_{\text{sed}} > mv_{\text{sed}}$, the variations seen by the particles have longer periods than those seen by air parcels or by particles that would fall through a stationary wave field with the same vertical structure. This is due to the fact that the particles travel in the same direction as the wave phase. In particular, when $c_{\phi_z} \simeq -v_{\text{sed}}$ (for $t \ll \left|\frac{1}{\omega + mv_{\text{sed}}}\right|$), one has:

$$Z(t) \simeq (W\cos(\phi_0) - v_{\text{sed}})t + Z_0 \tag{6}$$

and thus the wave can bring a significant contribution to the displacement of the crystal, a contribution which does not cancel out after one wave period. Such a configuration could also significantly modify the lifetime of the ice crystals, which could stay longer or shorter times in saturated or supersaturated regions than if only sedimentation were moving them. Since we only consider non-breaking waves, it should be noted, however, that the vertical wind contribution cannot overcome sedimentation in the long-term because stability requirements impose that $W < |c_{\phi_z}| \simeq |v_{\text{sed}}|$ (see appendix A). Furthermore, Eq. 6 implies a significant contribution of the wave to the motion of an individual ice crystal, but not necessarily an average effect on the ice crystal population. Indeed, if there is no preferential location of the crystals in the wave phases $\phi_0$ then some crystals are accelerated but others are slowed down.

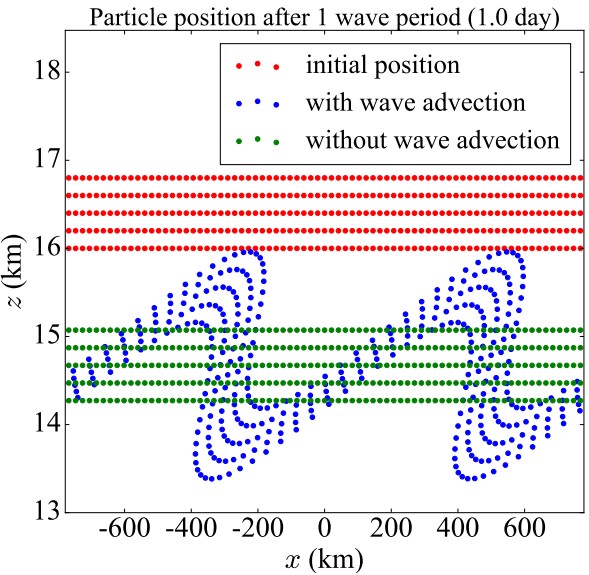

**Figure 1.** Evolution of the positions of sedimenting particles (initial positions in red) being advected by the wind field induced by a monochromatic wave (blue dots) or not (green dots) during one wave period. See text for a full description of the wave characteristics. Also note that the vertical scale has been enhanced by a factor of O(200) compared to the horizontal one.

For illustration, we show the effect of a monochromatic wave on the vertical motion of falling particles in a special configuration in Fig. 1. We assume that the particles fall at a constant speed of 2 cm/s (equivalent to a $\simeq 20$ $\mu$m-diameter spherical ice particle at $T = 190$ K and $P \simeq 120$ hPa). The wave is chosen with a period $T = \frac{2\pi}{\omega} \simeq 1$ day, a shallow vertical wavelength $\lambda_z = \frac{2\pi}{m} = 4$ km. The chosen squared Brunt-Väisälä frequency is $\bar{N}^2 = 2 \cdot 10^{-4}$ rad$^2$/s$^2$, i.e. intermediate between the stratosphere (where $\bar{N}^2 = 4 \cdot 10^{-4}$ rad$^2$/s$^2$ typically but can be as large as $1 \cdot 10^{-3}$ rad$^2$/s$^2$) and the troposphere (where typically $\bar{N}^2 = 1 \cdot 10^{-4}$ rad$^2$/s$^2$ but can be smaller near the bottom of the TTL) as expected for the transition region of the TTL. The horizontal wavelength is prescribed from the dispersion relation: $\lambda_x = \frac{2\pi}{k} = \frac{2\pi \bar{N}}{m\omega} \simeq 800$ km. The Eulerian temperature

perturbation has an amplitude $A_T = 1$ K, so that the vertical velocity amplitude is equal to

$$W = \frac{g}{\bar{N}^2}\omega\frac{A_T}{\bar{T}} \simeq 2 \text{ cm/s} \tag{7}$$

with $g = 9.81 \text{ m s}^{-2}$, $\bar{T} = 185$ K, while $U = \frac{|m|}{k}W \simeq 3.7$ m/s. Overall, the chosen wave characteristics are similar to those of equatorial waves commonly observed in radiosondes (e.g. Kim and Alexander, 2015). Although the only requirement for a

significant effect on the particle's speed is that the fall speed of the particle is close to the vertical phase speed of the wave, the integrated effect on the particle's displacement will be larger for low frequency waves, such as the one chosen in this example.

Figure 1 shows the initial positions of the particles (red dots) and their positions after one wave period (blue dots), so that all meteorological fields have the same value as at the beginning and the air parcels have returned to their initial positions. However, the particles have descended in altitude due to sedimentation. Without the wave, the particles would just fall to the

green positions. Due to the presence of the wave, advection by the vertical and horizontal winds significantly disperses the particles vertically relative to the no-wave case. Although we did *not* select the wave characteristics other than the intrinsic frequency to obtain it, a significant impact can be expected. We also checked that the monochromatic wave was stable (see appendix A).

Consistent with Eq. 6, Figure 1 shows that if the particles are of constant size with $v_{\text{sed}} \simeq c_{\phi_z}$, there is a significant impact

on the motion of individual particles. However, if the particles are initialized in all phases of the wave, there is no effect on the mean downward transport of the particles' population. The main impact is to disperse the particles vertically with some particles falling more slowly but others falling more rapidly when the wave is present. *Which of the two (increased or suppressed fall of the particles) will prevail in a realistic setting (i.e. including growth and sublimation)?*

## 2.2 Growing and sublimating ice crystals and wave-driven localization

### 2.2.1 Governing equations

Now that we have explored the impact of wave advection on particle transport, we turn to the case of ice crystals which can grow and sublimate, exchanging water molecules with their environment. We will however not consider the effect of the ice crystals on the relative humidity, equivalent to assuming that few of them are present. Consistently, we will neglect ice crystal aggregation; diffusional growth and sedimentation are thus the only microphysical processes active in our set up.

For crystals with a spherical shape, the rate of growth of the radius $r$ is given by Pruppacher and Klett (1978):

$$\frac{dr}{dt} = \frac{G'(r,T;\alpha_d)}{r}(RH_i - 1) \tag{8}$$

with $RH_i = \frac{q}{q_{\text{sat}}}$ the relative humidity with respect to ice ($q$ being the volume water vapor mixing ratio and $q_{\text{sat}}$ the volume saturation mixing ratio with respect to ice) and $G'(r,T;\alpha_d)$ the growth factor, a function of temperature $T$, the crystal radius $r$ and the deposition coefficient $\alpha_d$, given by:

$$G'(r,T) = \frac{1}{\rho_{\text{ice}}\left(\frac{R_v T}{e_{\text{sat}}(T)D'_v(r,T;\alpha_d)} + \frac{L_s}{Tk'_a(r,T)}\left(\frac{L_s}{TR_v} - 1\right)\right)}, \tag{9}$$

where $\rho_{\text{ice}} = 918$ kg/m$^3$ is the density of ice, $D_v'$ is the modified diffusivity of water vapor in air, $k_a'$ is the modified thermal conductivity of air, $e_{\text{sat}}$ the saturation water vapor pressure, $R_v = 462$ J/K/kg is the gas constant for water vapor and $L_s = 2.844 \cdot 10^6$ J/kg is the latent heat of sublimation of ice. The modified diffusivity $D_v'$ can be expressed as the product of the diffusivity $D_v(T)$ and the ventilation coefficients $f_{d,v}(r,T)$ (for large ice crystals) and non equilibrium corrections $f_{d,k}(r,T;\alpha_d)$ (for small ice crystals). Similarly, the modified thermal conductivity $k_a'$ is the product the conductivity $k_a(T)$ and the coefficients $f_{k,v}(r,T)$ and $f_{k,k}(r,T)$:

$$D_v' = D_v(T)f_{d,v}(r,T)f_{d,k}(r,T;\alpha_d),$$
$$k_a' = k_a(T)f_{k,v}(r,T)f_{k,k}(r,T). \tag{10}$$

Expressions for the $f$ coefficients can be found in Pruppacher and Klett (1978). For intermediate crystal sizes ($r \simeq 5$ to $50$ $\mu$m) and large deposition coefficients ($\alpha_d \geq 0.5$), $D_v' \simeq D_v(T)$ and $k_a' \simeq k_a(T)$, in which case $G'$ also is a function of $T$ only. The exact value of the deposition coefficient $\alpha_d$, which represents the fraction of water molecules colliding with the ice surface that effectively get incorporated into the ice crystal lattice, is not known. It could well vary with supersaturation or temperature and take any value from 0.001 to 1 and experimental studies have not been very helpful in constraining it so far (e.g. Magee et al., 2006; Skrotzki et al., 2013). However, atmospheric cloud observations are hard to reconcile with $\alpha_d$ values smaller than about 0.5 (e.g. Kärcher and Lohmann, 2002; Kay and Wood, 2008), and the recent discussion by Skrotzki et al. (2013) also recommends $0.2 \leq \alpha_d \leq 1$. We assume $\alpha_d = 0.5$, consistent with that literature. We note that the results presented below are not sensitive to the precise value of $\alpha_d$ as long as it is sufficiently large (larger than $\sim 0.5$), since $G'$ is then essentially a function of temperature only. However, for smaller values of $\alpha_d$, $G'$ depends both on $r$ (below some size up to tens of microns) and $\alpha_d$, and the results may be significantly altered both quantitatively and qualitatively.

In the following, we introduce $G = 2G'$, the rate of growth of the squared radius. The governing equations for the evolution of the ice crystal size and position in the monochromatic wave field are then:

$$\frac{dX}{dt} = U\cos(kX + mZ - \omega t + \phi),$$
$$\frac{dZ}{dt} = W\cos(kX + mZ - \omega t + \phi) - v_{\text{sed}}(r), \tag{11}$$
$$\frac{dr^2}{dt} = G(r, T(X,Z,t); \alpha_d)(RH_i(X,Z,t) - 1),$$

where again $X(t)$ and $Z(t)$ and the particle's horizontal and vertical positions. Of course, the trajectories of the crystals are irreversibly stopped when they fully sublimate ($r = 0$).

To solve these equations, we need to know the temperature and relative humidity at the position $X(t), Z(t)$ of the ice crystals. For the temperature, we could have used the polarization relations (Fritts and Alexander, 2003), but it is actually more relevant to derive $T$ and $RH_i$ directly by considering the field of vertical displacement induced by wave. Since we assume a monochromatic wave, the temperature $T(x,z,t)$ at any fixed (Eulerian) position $(x,z)$ in space is that of the air parcel that has

been adiabatically displaced to $(x, z)$ by the wave. Noting $Z_{\text{wave}}(x, z, t)$ the vertical component of this displacement, we have

$$T(x, z, t) = \underbrace{-\frac{g}{C_p} Z_{\text{wave}}(x, z, t)}_{\Delta T_{\text{wave}}} + \bar{T}(\bar{Z}(x, z, t)) \tag{12}$$

where $\bar{T}$ is the undisturbed (background) temperature at the air parcel equilibrium altitude $\bar{Z}$. Hence:

$$\bar{Z}(x, z, t) = z - Z_{\text{wave}}(x, z, t). \tag{13}$$

In the formulas above, the vertical displacement is given by:

$$Z_{\text{wave}}(x, z, t) = -\frac{W}{\omega} \sin(kx + mz - \omega t + \phi) \tag{14}$$

Regarding the pressure $P$, which is required in the relative humidity to evaluate the volume saturation mixing ratio $q_{\text{sat}} = \frac{e_{\text{sat}}(T)}{P}$, we assume hydrostatic equilibrium in the reference state $\bar{P}$ and neglect the pressure perturbations induced by the wave: $P(x, z, t) = \bar{P}(z)$. (Note that this is consistent with Eq. 12 which neglects the contribution of Eulerian pressure anomalies to

temperature changes.)

Regarding the water vapor mixing ratio, as mentioned above, it is estimated assuming that the crystals do *not* significantly deplete the water vapor content, which is then conserved. This last assumption will allow to reveal the wave-sedimentation interactions more clearly. It is valid when the ice crystal number concentrations are small, which is common in thin TTL cirrus (Krämer et al., 2009; Jensen et al., 2013), sometimes also referred to as Ultrathin Tropical Tropopause Clouds (UTTCs, Peter

et al., 2003). Indeed, Krämer et al. (2009) have argued that the relaxation time for supersaturation in those thin TTL clouds could be larger than a few hours. Furthermore, we assume that in the reference state the water vapor mixing ratio depends only on the vertical position, so that the water vapor mixing ratio at any position and time is given by:

$$q(x, z, t) = \bar{q}(\bar{Z}(x, z, t)) \tag{15}$$

For the reference-state $\bar{q}$ profile, we choose to keep a constant relative humidity with altitude, $RH_{i_c}$, i.e. :

$$\bar{q}(z) = RH_{i_c} q_{\text{sat}}(\bar{T}(z), \bar{P}(z)) \tag{16}$$

This set up retains the main characteristic of water vapor variability in the TTL, i.e. its decrease with altitude due to the decrease in saturation vapor pressure.

Finally, for the sedimentation speed $v_{\text{sed}}$ for spherical ice crystals, we will use the formulas provided in Pruppacher and Klett (1978), which are based on dimensional analysis and experiments, to evaluate the Reynolds number and then deduce the fall

velocity of the particles. For particles'radii between about 5 and 100 $\mu$m, those formulas give results close to the one obtained assuming Stokes'flow (within 8%), i.e.:

$$v_{\text{sed}} = \underbrace{\frac{2}{9} \frac{\rho_{\text{ice}} g}{\mu}}_{\alpha_{\text{sed}(\bar{T})}} r^2. \tag{17}$$

where $\mu(T)$ is the dynamic viscosity of air. However, for smaller particles, the formulas take into account Cunningham correction, which corrects for the non-continuum nature of the fluid for small particles. At the other end of the size spectrum, for particles larger than about 100 $\mu$m, the Stokes flow hypothesis is no longer valid and Eq. 17 overestimates the sedimentation speed, which is partly corrected in the formulas provided in Pruppacher and Klett (1978). We note also that those can be extended to non spherical particles (e.g. Mitchell, 1996; Heymsfield and Westbrook, 2010), but this will not be used in the simulations since we here consider spherical crystals in the growth calculations. Observations also suggest that most of the smallest ice crystals in TTL cirrus are quasi-spherical (McFarquhar et al., 2000); Lawson et al. (2008) found that most of the crystals with maximum dimension ($2r$) below 65 $\mu$m were quasi-spherical. However, the crystals above 65 $\mu$m observed by Lawson et al. (2008) mostly had aspherical shapes (including hexagonal and disk like). The dependency of the fall speed and therefore the mass flux on the particle shape is an important factor to take into account, and this will be discussed in Sect. 4.2.

### 2.2.2 System analysis

The previous set of equations will be the base for the simulations presented in Sect. 3.1. However, in order to analyze the dynamics of the system theoretically, we here consider three additional simplifications. First, we linearize the relative humidity term to only retain the essential oscillatory behavior (a strong, yet enlightening assumption given the non linearity of the saturation vapor pressure):

$$RH_{i_{\text{wave}}}(x,z,t) \simeq RH_{i_c}\left(1 - \underbrace{\frac{L_s}{R_v\bar{T}^2}\Delta T_{\text{wave}}(x,z,t)}_{\text{Clausius-Clapeyron term}} - \underbrace{\frac{g}{R\bar{T}}Z_{\text{wave}}(x,z,t)}_{\text{Pressure term}}\right) = RH_{i_c} + RH_{i_c}\underbrace{\left(\frac{L_s}{R_v\bar{T}^2}\frac{g}{C_p} - \frac{g}{R\bar{T}}\right)}_{\beta_G>0,\text{ only dependent on }\bar{T}}Z_{\text{wave}}(x,z,t)$$

(18)

Second, we keep only the dependence on $\bar{T}$ in $G$. This might induce some quantitative change, but the qualitative impact will be marginal.

Third, we use for $v_{\text{sed}}$ the simplified Stokes flow hypothesis from Eq. 17. This expression disregards corrections for small particles and large particles, but simplifies the algebra, since $r^2$ intervenes in that expression and in the growth expression.

Under those additional approximations, the system of equations (11) can be rearranged into two ordinary differential equations for the wave phase $\Psi = kX + mZ - \omega t + \phi$ along the ice crystal trajectory and for the squared radius of the particle $r^2$, namely:

$$\begin{cases} \dfrac{\mathrm{d}\Psi}{\mathrm{d}t} = -(\omega + m\alpha_{\text{sed}}r^2) \\ \dfrac{\mathrm{d}r^2}{\mathrm{d}t} = G\left(-\dfrac{W}{\omega}\beta_G\,RH_{i_c}\sin(\Psi) + RH_{i_c} - 1\right) \end{cases}$$

(19)

or, in a more abstract form:

$$
\begin{cases}
\dfrac{\mathrm{d}\Psi}{\mathrm{d}t} = -\underbrace{m\alpha_{\mathrm{sed}}\,r^2}_{A} - \underbrace{\omega}_{B}, \\[4mm]
\dfrac{\mathrm{d}r^2}{\mathrm{d}t} = \underbrace{-G\,\dfrac{W}{\omega}\,\beta_G\,RH_{i_c}\,\sin\left(\Psi\right)}_{C} + \underbrace{G\left(RH_{i_c}-1\right)}_{D}
\end{cases}
\tag{20}
$$

This set of equations reveals the simple properties of the system studied. It can be seen that the system has fixed points, characterized by $\frac{\mathrm{d}\Psi}{\mathrm{d}t}=0$ and $\frac{\mathrm{d}r^2}{\mathrm{d}t}=0$, provided that the constraints:

5  $\quad \dfrac{B}{A} < 0$, i.e. $c_{\phi_z} = \dfrac{\omega}{m} < 0$, which corresponds to upward propagating wave packets $\hfill$ (21)

and

$$
\left|\frac{D}{C}\right| \le 1 \text{ which ensures the existence of regions of } RH_{i_{\mathrm{wave}}} = 100\% \tag{22}
$$

are satisfied. Then, there are 1 or 2 (depending if it is an equality or a strict inequality in the previous equation) fixed points, which are given by:

$$
\begin{cases}
r_f^2 = -\dfrac{B}{A} = -\dfrac{c_{\phi_z}}{\alpha_{\mathrm{sed}}}, \\[4mm]
\sin\left(\Psi_f\right) = -\dfrac{D}{C} = \dfrac{\omega}{W\beta_G}\,\dfrac{RH_{ic}-1}{RH_{ic}}.
\end{cases}
\tag{23}
$$

The first equation states that the sedimentation speed at the fixed point is equal to the wave vertical phase speed ($v_{\mathrm{sed}}(r_f) = c_{\phi_z}$). The second equation states that $RH_{i_{\mathrm{wave}}}(\Psi_f) = 100\%$, i.e. the fixed points are located where the environment is exactly at saturation so that the ice crystals' radius is constant (this is also true in the full system, as can be seen in the last equation of System 11). Note that the second equation in (23) corresponds to two possible fixed points in the wave phase space, since

15  $\quad \sin\left(\Psi_f\right) = \sin\left(\pi - \Psi_f\right)$.

To gain further insights into the behavior of the trajectories in the vicinity of the fixed points, it is common to examine the linearized system. The Jacobian matrix at the fixed points is:

$$
J = \begin{pmatrix} 0 & -A \\ D\cos\left(\Psi_f\right) & 0 \end{pmatrix}, \tag{24}
$$

and its eigenvalues $\lambda$ verify:

20  $\quad \lambda^2 = -A\,D\cos\left(\Psi_f\right) = \dfrac{W}{c_{\phi_z}}\alpha_{\mathrm{sed}}\beta_G G\,RH_{i_c}\cos\left(\Psi_f\right) = \pm\dfrac{W}{c_{\phi_z}}\alpha_{\mathrm{sed}}\beta_G G\,RH_{i_c}\sqrt{1-\left(\dfrac{RH_{i_c}-1}{RH_{i_c}}\dfrac{\omega}{W\beta_G}\right)^2}.$  (25)

with the plus sign corresponding to $\Psi_f$ in $\left[-\frac{\pi}{2}, \frac{\pi}{2}\right]$.

Given those eigenvalues, the two fixed points of the system deduced from the theoretical analysis are:

- a *saddle point*, for which $\lambda^2 > 0$ and the two eigenvalues are reals of opposite signs. The value of the phase at the saddle point $\Psi_s$ is characterized by $W\cos(\Psi_s) < 0$ (given that $c_{\phi_z} < 0$), so that the vertical wind anomaly induced by the wave $w'$ is negative. The particles initially around that fixed point move away from it with hyperbolic trajectories in the $\Psi - r$ space.

- an *elliptic point*, for which $\lambda^2 < 0$ and the eigenvalues are both purely imaginary and conjugate to each other. This fixed point is located in the cooling phase of the wave, i.e. its phase $\Psi_e$ is characterized by a positive vertical wind anomaly: $w' = W\cos(\Psi_e)$ is positive. In the fully linear system one would expect periodic trajectories in the neighborhood of the elliptic point, with particles cycling periodically around it, at a frequency:

$$\omega_{cx} \simeq \sqrt{-\frac{W}{c_{\phi_z}}\alpha_{\text{sed}}\beta_G G\, RH_{i_c}\sqrt{1 - \left(\frac{RH_{i_c}-1}{RH_{i_c}}\frac{\omega}{W\beta_G}\right)^2}} \tag{26}$$

However, since the real part of both eigenvalues is zero for this point, the linear analysis is actually not sufficient to conclude regarding the behavior of the trajectories in the non linear system (Hirsch et al., 2016).

The linear analysis above only brings qualitative insights on the system behavior near the saddle point, and cannot be applied rigorously near the elliptic point. However, the behavior of the system can still be studied further theoretically, since it turns out to be Hamiltonian (see Appendix B). The Hamiltonian function $H$ can be expressed as

$$H(r,\Psi) = \frac{A}{2}\left(r^2 + \frac{B}{A}\right)^2 - C\cos(\Psi) + D\,\Psi \tag{27}$$

The trajectories in the $r - \Psi$ space correspond to lines of constant $H$. This formulation makes it clear that there are periodic trajectories around the elliptic point, as long as $\left|\frac{D}{C}\right| < 1$ since in that case the elliptic point is a local extremum of $H$ (see Appendix B).

Figure 2 shows the phase portrait of the system (trajectories of the particles in the $\Psi - r$ phase space, i.e. contours of constant $H$) for different background relative humidity $RH_{i_c}$. On this figure, only the part of the trajectories corresponding to $r \geq 0$ are shown, since for $r = 0$ the crystals have fully sublimated and $r < 0$ has no physical meaning. The characteristics of the wave and of the background state that have been used in this figure are summarized in Table 1; unless stated otherwise, they will also be assumed in the remainder of the study. Although the wave temperature amplitude may seem large (1.7 K), they correspond to the large events that regularly occur in the TTL, as observed by Kim and Alexander (2013). In particular, one specific wave event observed near Guam during ATTREX was observed to induce larger zonal wind and temperature fluctuations, with a comparable period to the wave chosen here (Kim (2015), also see the case-study in Podglajen et al. (2017)).

Figure 2 emphasizes the existence of closed (periodic) orbits near the elliptic fixed points, located where $RH_i = 100\%$ in the cooling phase of the wave and delimited by the red lines. The crystals in that region remain in a specific phase of the wave near the elliptic point. We call this behavior of the crystals remaining in a specific wave phase the *wave-driven localization* of ice crystals. Furthermore, the figure emphasizes that crystals initially placed in the negative temperature anomaly region will tend

**Table 1.** Wave and background state characteristics assumed for the sedimentation-growth simulations reported in this section. $\bar{P}$: average pressure (for the simplified system), $\bar{T}$: average temperature (for the simplified system), $\bar{N}$: background Brunt-Väisälä frequency, $\frac{2\pi}{\omega}$: wave period, $\lambda_z$ wave vertical wavelength, $A_T$: wave (Eulerian) temperature T amplitude, $W$: wave vertical wind amplitude, $U$: wave zonal wind amplitude, $c_{\phi_z} = \frac{\omega}{m}$: wave vertical phase speed, $\frac{2\pi}{\omega_{cx}}$: period of the oscillations of the crystals around the elliptic point.

| $\bar{P}$ | $\bar{T}$ | $\bar{N}^2$ | $\frac{2\pi}{\omega}$ | $\lambda_z$ | $A_T$ | $W$ | $U$ | $c_{\phi_z}$ | $\frac{2\pi}{\omega_{cx}}$ $(RH_{i_c} = 0.85)$ |
|---|---|---|---|---|---|---|---|---|---|
| 120 hPa | 195 K | $2 \cdot 10^{-4}$ rad$^2$/s$^2$ | 2 days | 4 km | $\sim 1.7$ K | 1.57 cm/s | 6 m/s | $-2.3$ cm/s | $\sim$12 hours |

to grow and leave that region to move into the positive temperature region, spending some time on the way in the cooling phase of the wave. Even if they are outside the region of permanent trapping near the elliptic point, some trajectories remain near that region a significant amount of time, so that the preferential location of crystals due to the *wave-driven localization* might manifest itself for a fraction of the crystal population larger than that corresponding to the area enclosed by the red curves.

### 2.2.3 Physical understanding

The existence of the two fixed points can be easily understood physically. The elliptic point is located at $RH_i = 100\%$ and $\frac{\partial RH_i}{\partial z} > 0$. Hence, if ice crystals fall below that fixed point, they fall into subsaturated air ($RH_i < 100\%$) and sublimate, which reduces their mass and their fall velocity. They fall more slowly and are caught up again by the wave phase which is also descending. They may then be transported into supersaturated regions where they grow, increasing their weight and fall speed and moving them back into the equilibrium phase. The trajectory of an ice crystal in altitude-time space around the elliptic fixed point is sketched in Fig. 3, to illustrate the previous explanation. It can be noted that the ice crystal cycles around the elliptic point with a period of about 12 hours, consistent with Eq. (26) (see also Table 1). At the saddle fixed point, the reverse feedback is acting with subsaturated air above and supersaturated air below, so that the crystals move further away from this equilibrium point.

In many aspects, the mechanism presented here is similar to the stabilization mechanism proposed by Luo et al. (2003) to explain the existence of Ultrathin Tropical Tropopause Clouds. Those authors considered ice crystals in a stationary vertical wind and relative humidity profile, and neglected the horizontal wind shear so that only vertical motions were examined. Then, a system of equations similar to (19) can be derived to describe the evolution of the crystal radius and position, where essentially the altitude replaces $\Psi$ and the vertical wind replaces the vertical wave phase speed. In that framework, for the same reason explained above for the elliptic point, ice crystals are stabilized in regions where $RH_i = 100\%$, $\frac{\partial RH_i}{\partial z} > 0$ provided that the vertical wind is constant or decreasing with altitude. Taking a vertical profile, the location where the wave-driven localization occurs in our analysis (i.e., the elliptic point) is thus the same as the one where the stabilization effect of Luo et al. (2003) is expected ($RH_i = 100\%$, $\frac{\partial RH_i}{\partial z} > 0$). However, since waves essentially dominate the variability of the vertical wind in the TTL, the setting of a quasi-monochromatic wave considered above is likely more realistic than the stationary vertical wind profile without horizontal wind shear used by Luo et al. (2003).

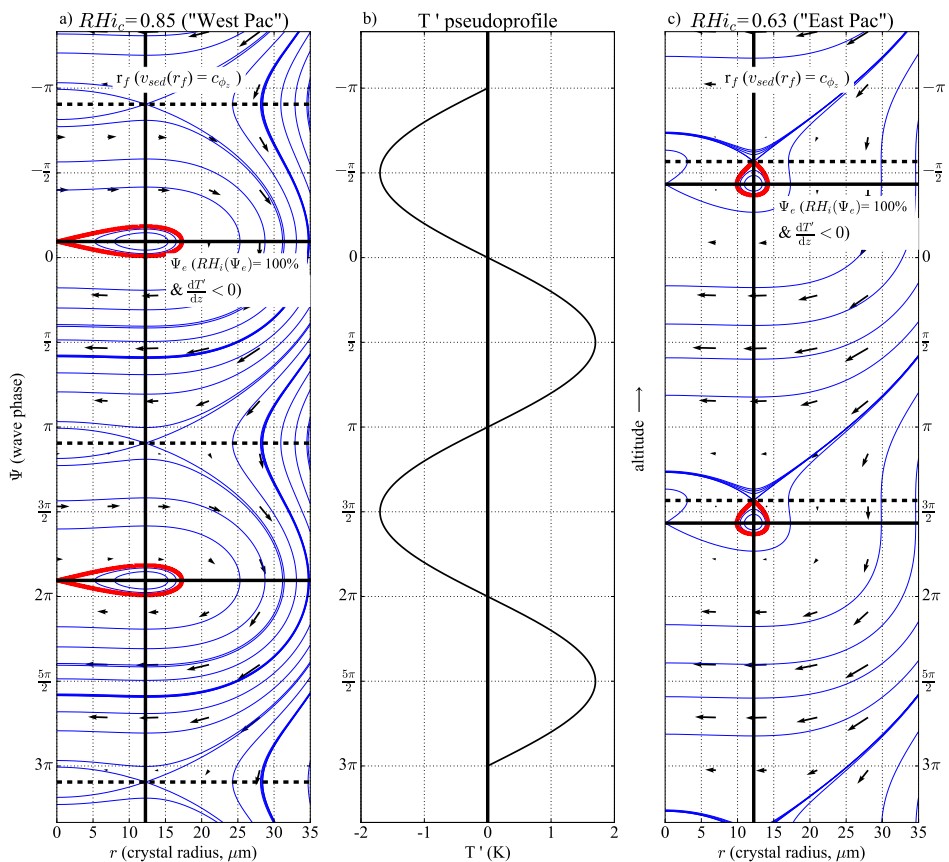

**Figure 2.** (Side panels) Representation in the $\Psi - r$ phase space (phase portrait) of ice crystals trajectories (blue lines) obtained with wave parameters given in Table 1 for two different background relative humidities: a moist case similar to the western Pacific ($RH_{ic} = 0.85$, panel a)) and a drier case similar to the eastern Pacific ($RH_{ic} = 0.63$, panel c)). These trajectories correspond to constant values of the Hamiltonian function (Eq. (27)). Black arrows along the blue lines indicate the trajectory direction. The vertical heavy line corresponds to $r_f$, the particle radius at the fixed points. The heavy horizontal black line show the location of the wave phase of the elliptic (solid) and saddle (dashed) fixed points. The heavy red lines limit the area around the elliptic fixed point where the crystals have "perpetual" periodic trajectories. It should be noted that the y-axis ($\Psi$) is similar to a vertical profile, with decreasing $\Psi$ corresponding to increasing altitude (since $m < 0$); the corresponding temperature anomaly profile due to the wave is sketched on panel b).

Returning to the location of the elliptic point where the crystals are localized, it should furthermore be noted that, given the tendency of some of the crystals to stay near the thermodynamic equilibrium phase of the wave (i.e. $RH_i = 100\%$), there is for those a net impact of the wave on their vertical displacement. Indeed, at the elliptic fixed point, the crystals will endure a constant wave-induced horizontal velocity $U \cos(\Psi_f)$ and a constant vertical velocity $W \cos(\Psi_f)$. Since the elliptic fixed point is the one for which $W \cos(\Psi_f) > 0$, the consequence is that sedimentation is slowed down by wave advection.

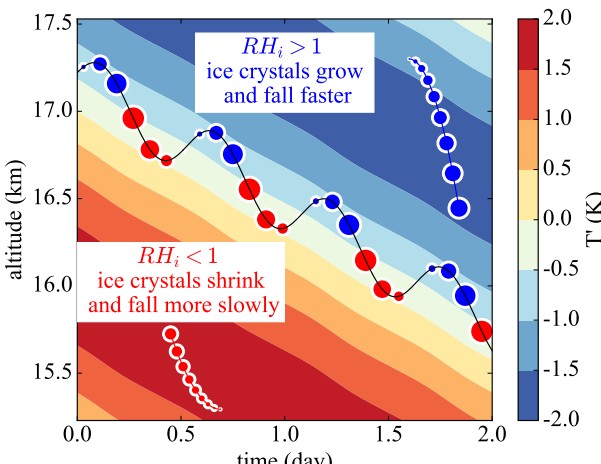

**Figure 3.** Representation in altitude-time space of an ice crystal trajectory in the moist ($RH_{ic} = 0.85$) simulations. The colors correspond to the temperature anomaly profiles induced by the wave at the $X$ position of the crystal, which also correspond to relative humidity anomalies. It is important to note that as a consequence of the crystal horizontal motion the $X$ position at which the profiles are taken changes with time. The black line corresponds to the ice crystal trajectory and the circular markers represent the ice crystal size. Blue circles indicate growing ice crystals ($RH_i > 1$) whereas red circles indicate sublimating ice crystals. Idealized trajectories of growing (blue) and sublimating (red) crystals in constant $RH_i$ backgrounds are also shown for a pedagogic purpose in the lower left and upper right corners.

## 2.3 Sensitivity of the wave-driven localization

### 2.3.1 Moist versus dry environments

Figure 2 displays crystals trajectories for two background relative humidities, a moist ($RH_{i_c} = 85\%$) and a dry ($RH_{i_c} = 63\%$) scenario. These two scenarios serve to provide an explanation for geographical differences in TTL cirrus clouds over the Pacific ocean recently reported by Kim et al. (2016). Using in situ observations, their study examined the relationship between TTL cirrus clouds and temperature anomalies during boreal winter time, and found different relations between the tropical eastern and western Pacific. In the eastern Pacific, TTL cirrus are tied to the cold phases of the waves (the minimum of temperature anomaly $T'$ in Fig. (2), central panel). In the western Pacific upper TTL, cirrus are more frequent in the negative vertical temperature gradient phase ($dT'/dz < 0$; the cooling phase of the waves if they propagate upward). This difference between the eastern and the western Pacific is probably not due to differences in wave amplitudes or characteristics: indeed, observations suggest that the lower frequency waves responsible for the temperature fluctuations do not show a strong geographic variability within the equatorial region (Podglajen et al., 2016a; Kim et al., 2016).

Another reason could be the different mean relative humidities between the two regions, the convective western Pacific being moister on average than the dry eastern Pacific. Two different explanations relying on the different background humidities could then explain the observations. The first, proposed by Kim et al. (2016), is that in a moist environment smaller temperature

perturbations are required to go over the supersaturation threshold for ice nucleation than in a dry environment. Hence, ice formation may happen in the cooling phase of the wave where temperature perturbations are smaller in the moist regions while only the coldest wave phases provide sufficient supersaturation for nucleation in the dry eastern Pacific. However, this explanation does not take into account the life cycle of the ice crystals. Figure 2 illustrates an alternative explanation based on the interaction of sedimentation and growth with the wave field: in the dry environment, the elliptic fixed point is located near the cold phase of the wave ($T' < 0$) while in the moist environment it is in the cooling phase of the wave ($dT'/dz < 0$). The wave-driven localization effect may therefore be responsible for the preferential location of ice crystals and clouds in specific phases of wave-induced disturbances, like those reported by Kim et al. (2016).

### 2.3.2   Sensitivity to wave parameters

Besides the strong sensitivity to the background relative humidity, it is also interesting to investigate qualitatively the sensitivity of the wave-driven localization to the wave parameters. One metric to quantify the relevance of wave-driven localization is the fraction of phase space $\mathcal{F}_p$ in which ice crystals of radius $r = r_f$ are affected by that process, i.e. $\mathcal{F}_p = \Delta\Psi/(2\pi)$ where $\Delta\Psi$ is the phase difference of the two points limiting the closed orbits region at $r = r_f$.

From Appendix B and by looking at Fig. 2, it appears that the trajectory passing through the saddle point delimits the closed orbits, as long as it does not intersect the line $r = 0$. In the latter case, it is the trajectory passing through the point $r = 0$, $\Psi = \Psi_e$ ($\Psi_e$ being the location of the elliptic point) which is the limit of the region with periodic orbits. The mathematical constraints are detailed in Appendix B and the corresponding limits of the regions with closed orbits for the two relative humidities are shown by the red curves in Fig. 2.

The dependency of $\mathcal{F}_p$ to wave parameters is illustrated in Fig. 4 for $RH_{ic} = 0.85$. This figure shows the existence of two regimes. In the first regime (wave periods below 1 day in that case), $\mathcal{F}_p$ depends on the distance between the elliptic and saddle fixed points, and thus increases with the temperature amplitude of the wave $\frac{W}{\omega}$ (see Eq. (23)). $\mathcal{F}_p$ thus increases with the wave period $T = 2\pi/\omega$ (it would also increase with with $W$), with no dependency on the vertical wavelength $\lambda_z = 2\pi/m$. This happens until the second regime is reached for larger periods. In this regime, $\mathcal{F}_p$ is limited by crystals that fully sublimate. The wave parameters mainly act through modifying the radius at the fixed point (through $v_{\text{sed}}(r_f) = -\frac{\omega}{m} = -\frac{\lambda_z}{T}$), i.e. the position of the center of the orbits and their size. Hence, increasing $\lambda_z$ increases $\mathcal{F}_p$ mainly through an increase of $r_f$ while increasing $T$ decreases $\mathcal{F}_p$ mainly through both a decrease of $r_f$ and an increase of the size of the orbits in $r$.

This exercise shows that low frequency and large vertical wavelength waves are in general more susceptible to exhibit the wave-driven localization effect. However, it is quantitatively present ($> 5\%$ of the wave phase space) for a large fraction of wave parameters, and is not restricted to the parameters chosen above.

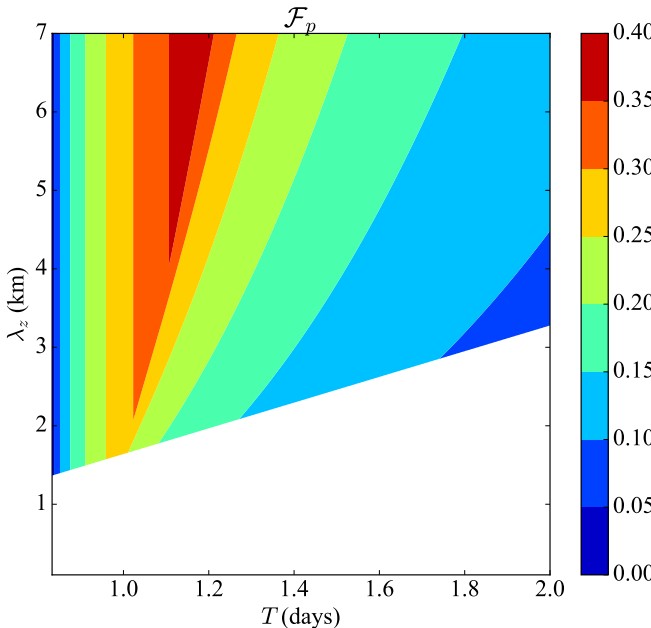

**Figure 4.** Fraction $\mathcal{F}_p$ of the wave phase space affected by the wave driven localization, as a function of the period $T = 2\pi/\omega$ and vertical wavelength $\lambda_z = 2\pi/m$ of the wave. $RH_{ic} = 0.85$ and the parameters other than $m$ and $\omega$ are given in Table 1. The regions of wave instability are not shown (white).

# 3 Wave advection impact in realistic settings and in observations

In the previous section, a simplified framework was introduced to investigate the potential effects of the wave on ice crystal motions. The present section aims at confirming those effects with more realistic numerical simulations and to investigate them in observations.

## 3.1 Full simulations

We now present numerical simulations of the full System (11). The set up is the one presented in Sect. 2.2.1 (constant background relative humidity and idealized wave), but without the additional simplifications in Sect. 2.2.2. The full equations are solved, and the dependency of the microphysical parameters on the varying background temperature, pressure and crystal size are included. We still assume that there are few ice crystals, i.e. there is no water consumption. The goal is to extend the analysis of the simplified system, to check whether the expected patterns of cloud occurrence in preferred wave phases appear, and to illustrate the impact of wave advection on transport. For that purpose, a population of ice crystals with radius 5 $\mu$m is initialized in all phases of the idealized wave. They then grow or sublimate depending on the environment relative humidity. Sublimated ice crystals are irremediably lost, while the others continue their trajectories down to the bottom of the domain.

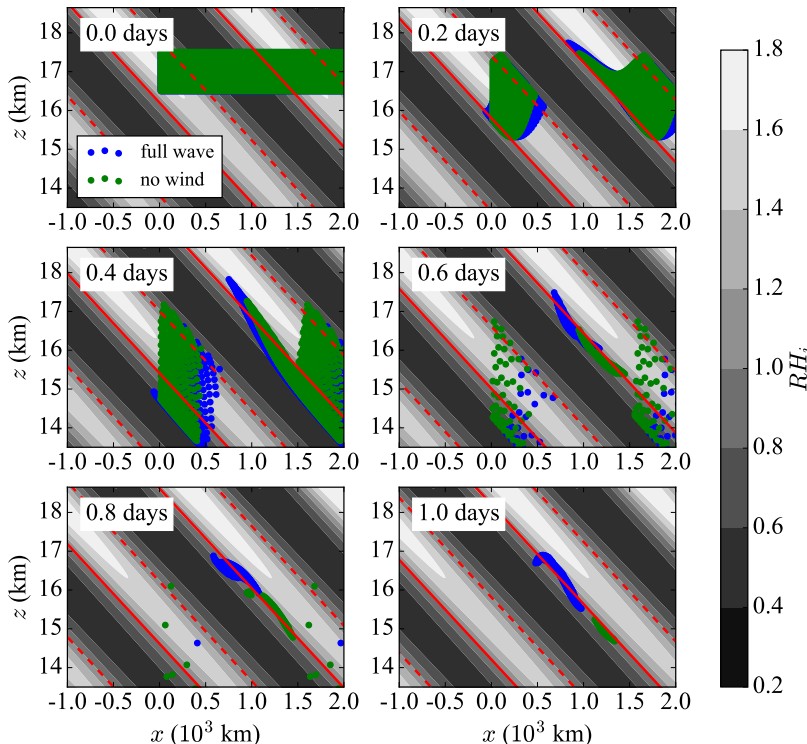

**Figure 5.** Simulations of ice crystal growth and sedimentation in an idealized wave field at $RH_{i_c} = 85\%$. Different integration times are displayed, up to half a wave period, i.e. one day. The black and white background represents the $RH_i$ spatial variability and the colored dots correspond to the position of the ice crystals. The different colors correspond to simulations with different processes (un)accounted for: the blue dots represent the ice crystals' positions for the full simulation (wave advection and temperature fluctuations) whereas for the green dots both the horizontal and vertical winds induced by the wave have been neglected. The solid and dashed red lines respectively correspond to the locations of the elliptic and saddle points; for both of those $RH_i = 100\%$. The initial ice crystal radius used for all crystals is 5 $\mu$m.

Figure 5 displays the results of a simulation at $RH_{i_c} = 85\%$, integrated during one day. The ice crystals are initialized with a radius of 5 $\mu$m around 17 km (16.5 to 17.5 km) in regularly gridded horizontal positions spanning all the phases of the wave. The different colors correspond to simulations accounting for wave advection or not, and will be discussed later.

Consistent with the analysis of the simplified system in Sect. 2 (see Fig. 2), three types of ice crystal behaviors are evident in Fig. 5:

- Ice crystals initialized in (highly) subsaturated regions ($RH_i < 70\%$, black in Fig. 5) quickly sublimate.

- Ice crystals initialized in supersaturated regions below the saddle point grow and fall, cross the $RH_i = 100\%$ region (the elliptic fixed point) to sublimate below it, about half a vertical wavelength below their initial position.

– Ice crystals initialized near the elliptic fixed point ($RH_i \sim 100\%$) tend to stay in the same wave phase and fall more slowly. This last group of ice crystals, which in this case amounts for about $5\%$ of the initialized crystals, tends to stay in the TTL a significant time after their initialization and may then be more frequently sampled during observations.

The qualitative behavior between the full ice crystal trajectories and the simplified system are thus similar. Now, *is the remaining ice crystal population sensitive to the background relative humidity?*

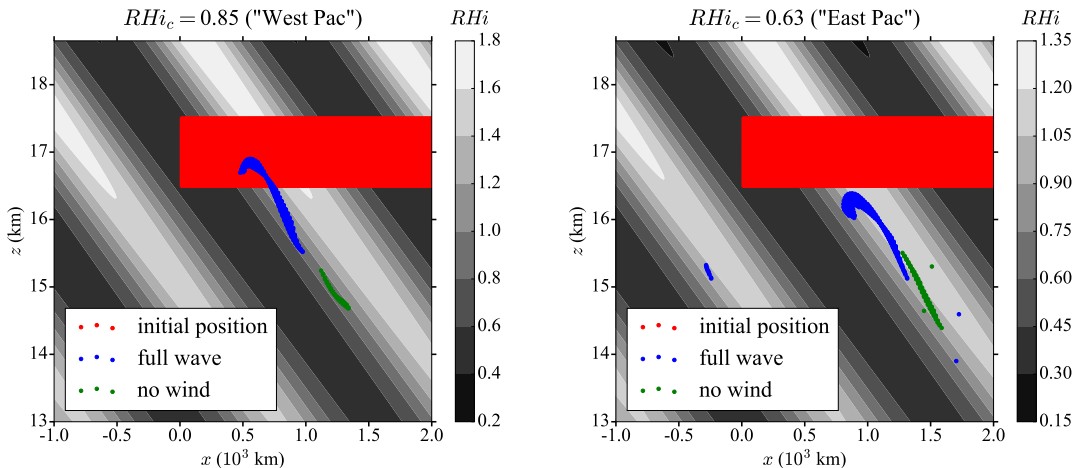

**Figure 6.** Simulations of ice crystal growth and sedimentation in an idealized wave field at $RH_{i_c} = 85\%$ (left) and $RH_{i_c} = 63\%$ (right), after half a wave period, i.e. one day. Note that the scale for the relative humidity (background in black and white) differs between the two panels. The left panel is similar to the bottom right panel of Fig. 5. Dots represent the positions of the ice crystals, with the different colors corresponding to different trajectory simulations: the red dots are the initial positions of the particles, the blue dots represent the ice crystals' positions for the full simulation (wave advection and temperature fluctuations) whereas the green dots are ice crystals for which the wave-induced wind has been neglected in the simulation. The initial ice crystal radius used for all crystals is 5 $\mu$m. Note how the remaining crystals tend to be regrouped in the phase of the wave with $RH_i$=100%.

In Fig. 6, the results of the previous simulation ($RH_{i_c} = 85\%$) are compared with the dry case ($RH_{i_c} = 63\%$) after one day of integration, i.e. half a wave period. The end position of the crystals are shown by the blue dots. The main characteristics described in Sect. 2.3.1 can again be noticed, with the remaining ice crystals (about 5 % of the initialized crystals in both cases) preferentially encountered where the relative humidity is near $100\%$ and in the $\partial T'/\partial z < 0$ ($\partial RH_i/\partial z > 0$) phase of the wave. We also note that, in the dry case (right panel of Fig. 6), there are remaining crystals after half a wave period even though those were not initialized in the region of perpetual oscillations expected from the theoretical analysis (their radius of 5 $\mu$m being too small, see panel c) of Fig. 2). They are however sufficiently close to this point to remain near the elliptic point a significant amount of time. Thus, due to the presence of the wave, those ice crystals can survive longer and remain in the TTL even though the large scale state is subsaturated.

When the background relative humidity $RH_{i_c}$ is increased even further (e.g. $RH_{i_c} \geq 100\%$), the impact is similar with the remaining ice crystals near the elliptic point which will be located at a different position in phase such that $RH_i(\Psi_f) = 100\%$.

However, it should be noted that, for the ice crystals away from the elliptic point, the background relative humidity changes the relative proportion of ice crystals that sublimate versus those that quickly fall out of the TTL through the lower boundary of the domain.

In Sect. 2, we emphasized that ice crystals may encounter significant wave-advection when they are more frequent in specific wave phases. The feedback between sedimentation and growth tends to localize the remaining ice crystals preferentially in the specific wave phase where the elliptic point is located. Hence, we can expect a mean impact on the downward velocity of the ice crystals.

To show that more precisely, we have also performed simulations without the wave advection. The crystals just fall, seeing the relative humidity perturbations created by the wave, but not the wave-induced wind perturbations. The end position of the ice crystals for the different simulations are represented in Fig. 5 and Fig. 6 by the green dots. For the low relative humidity case ($RH_{i_c} = 63\%$), the differences in the positions of the remaining ice crystals between the simulations with and without the wave-induced wind is limited, which is expected since the phase where the elliptic point is located in that case is characterized by very small vertical wind anomalies $W \cos \Psi_e \ll W$. Yet, more crystals survive and do no sublimate in the full wind simulation than in the no-wind one ($\sim 5\%$ vs $1\%$). In the high relative humidity set up, the differences between the full simulations and the no wind simulations already noticed for low $RH_i$ are enhanced: the downward sedimentation velocity of the crystals has been slowed down by almost a factor of two, due to the crystals remaining in the phase $W \cos \Psi_e \simeq W$. This is in striking contrast with the simulation for which the wave wind was not accounted for.

The differences seen between these 2 simulations suggest that *wave advection slows down the descent of the ice crystals*. Indeed, when the full wave is appropriately accounted for, the crystals' downward motion at the elliptic point happens at a (negative) vertical speed:

$$w_{c_{full}} = v_{sed_{full}} + W \cos \Psi_e = c_{\phi_z} + W \cos \Psi_e \tag{28}$$

whereas if the advection by the wave-induced wind is neglected, the crystals fall more rapidly and their downward vertical speed becomes larger (more negative), specifically:

$$w_{c_{no\ hor}} = w_{c_{no\ wind}} = c_{\phi_z} < w_{c_{full}} (< 0). \tag{29}$$

This is due to the fact that the vertical wind anomaly $w' = W \cos \Psi_e$ is positive at the elliptic point. Thus, the wave advection can significantly slow down the fall of ice crystals (in our example, by a factor of nearly 2). It is interesting to note that this effect does not only come from the wave-induced vertical wind but also from the horizontal wind disturbance, as detailed in Appendix C. The contribution of the wave to both wind components is thus central to entirely apprehend its impacts on ice crystals.

## 3.2   Quantifying the impact of wave advection on the vertical transport using observations

Overall, the experiments described above show that advection by the wave-induced wind has an impact on the sedimentation-growth of ice crystals and can significantly diminish the sedimentation mass flux, which suggests a mean impact of wave

advection on the average dehydration efficiency of ice crystals. The downward water mass flux needed to close water budget of the TTL may then be significantly affected by the waves. However, those simulations remain idealized: for instance, the described wave-driven localization is tied to the initialization of the crystals in all phases of the wave, in particular in the phase where $RH_i \simeq 100\%$. Ice nucleation at low TTL temperatures is only active for $RH_i \simeq 120\%$ (heterogeneous nucleation) to

160% (homogeneous nucleation) so that small-scale gravity waves superimposed to the large-scale wave would be required to bring everywhere the air to the supersaturation threshold for nucleation. Given the number of additional assumptions and the complexity that would be required to represent them in a more complete setting, we leave those investigations for future work. Rather, to investigate the relevance of this potential effect to the atmosphere, we turn to aircraft observations in the TTL from the Airborne Tropical TRopopause EXperiment (ATTREX). In both 2013 and 2014 the Global Hawk carried a Fast

Cloud Droplet Probe (FCDP) which was used to measure the size distribution and concentration of ice crystals from 1 to 50 microns diameter, and measured at about 1 Hz (180 m horizontal resolution). Temperature and pressure were provided at 1 Hz by the NASA Ames Meteorological Measurement System (MMS). Finally, of interest for this study, a Microwave Temperature Profiler provided estimates of the local temperature lapse rate with a resolution of about 3 km along the flight track.

To evaluate the impact of the wave vertical wind on the ice mass flux, it would seem natural to use the standard altitude

coordinate. In that case, the average ice mass vertical flux $F_{\mathrm{ice}_z}$ through a constant-altitude surface, is expressed as:

$$F_{\mathrm{ice}_z} = \overline{w' q_{\mathrm{ice}}} - F_{\mathrm{sed}} \tag{30}$$

where $q_{\mathrm{ice}}$ is the ice mass concentration and $F_{\mathrm{sed}}$ the sedimentation flux, given by

$$F_{\mathrm{sed}} = \int\limits_{0}^{+\infty} m(D) N(D) v_{\mathrm{sed}}(D) \mathrm{d}D \tag{31}$$

with $m(D)$ the mass of an ice crystal of maximum dimension $D$, $N(D)$ the number density distribution and $v_{\mathrm{sed}}(D)$ the

sedimentation speed. Assuming spherical ice crystals, $D$ is the diameter and $m(D) = \rho_{\mathrm{ice}} \frac{\pi}{6} D^3$.

In practice, however, the method suggested by Eq. 30 turns out to be difficult to apply to observations, since vertical velocities are dominated by high frequency waves (Podglajen et al., 2016a) whose mean impact cancels out ($\overline{w' q_{\mathrm{ice}}}$=0) but which add noise to the observational estimate. To overcome these issues, we use isentropic coordinates in which reversible motions are filtered out. This acknowledges that efficient, irreversible dehydration only occurs when ice crystals cross isentropic surfaces.

Switching to isentropic coordinates $\theta$ on the vertical, the ice crystal vertical speed becomes:

$$v_\theta = v_{\mathrm{sed}} \frac{\partial \theta}{\partial z} \tag{32}$$

The wave advection impact is then hidden in the wave-induced stability fluctuations $\frac{\partial \theta'}{\partial z}$:

$$\frac{\partial \theta}{\partial z}(x,y,z,t) = \underbrace{\frac{\partial \bar{\theta}}{\partial z}(z)}_{\text{background}} + \underbrace{\frac{\partial \theta'}{\partial z}(x,y,z,t)}_{\text{wave}} \tag{33}$$

and this can be readily computed from observations of vertical temperature profiles and ice crystals size, such as those from

ATTREX campaign used by Kim et al. (2016). The results, presented in Appendix D, show generally smaller $\frac{\partial \theta'}{\partial z}$ within clouds and are consistent with Kim et al. (2016) which showed that within clouds lower $\frac{\partial T'}{\partial z}$ were found.

There is thus a systematic relation between clouds and anomalies of $\frac{\partial \theta'}{\partial z}$. This relation suggests that there is an impact of wave advection on the ice flux. To quantify this more precisely, we use a relevant quantity for dehydration in isentropic coordinate, the cross-isentropic vertical mass flux of total water $F_{H_2O_\theta}$:

$$F_{H_2O_\theta} = q_{H_2O} \frac{\dot{\theta}}{\frac{\partial \theta}{\partial z}} - F_{sed} = \frac{1}{\frac{\partial \theta}{\partial z}} \left[ q_{H_2O} \dot{\theta} - F_{sed} \frac{\partial \bar{\theta}}{\partial z} - F_{sed} \frac{\partial \theta'}{\partial z} \right] \tag{34}$$

with $q_{H_2O}$ the total water mass concentration. In the first part of the equality, the term proportional to the diabatic heating rate $\dot{\theta}$ corresponds to air masses crossing isentropes and transporting their water vapor and water condensates. The term $F_{sed}$ corresponds to the cross-isentropic flux due to sedimentation. We have here neglected eddy diffusive fluxes. *Permanent* dehydration due to cloud formation will be brought by this irreversible cross isentropic water mass flux $F_{H_2O_\theta}$. The second part of the equation splits the sedimentation term into two terms, in order to emphasize the modulation introduced by the waves through their stability impact. Strictly speaking, the impact is actually more on the radiatively driven mass flux than on the sedimentation flux. However, we apply this splitting for the purpose of illustration, since a direct estimate would require further information such as the radiative heating rates within the clouds observed during the campaign. Figure 7 represents the sedimentation flux for the whole ATTREX campaign, as well as for the eastern and western Pacific flights separately. It can be clearly seen that the downward flux due to sedimentation is stronger over the cold, convective western Pacific than over the eastern Pacific. Furthermore, the wave-advection impact is represented by the black curves for the whole campaign. The full curve corresponds to the term $\left| F_{sed} \frac{\partial \theta'}{\partial z} / \frac{\partial \theta}{\partial z} \right|$, which represents the wave advection impact on sedimentation. In the upper TTL, it is is on average about 10% of the mean downward flux. This influence is not very large, but still significant. The wave advection impact might hence be worth taking into account.

## 4 Discussion

### 4.1 An alternative mechanism to explain the clear-sky and cloudy air relative humidity in the UTLS?

The frequency distribution of $RH_i$ in the upper troposphere shows two robust characteristics: 1) the common occurrence of 100% relative humidity within clouds and 2) the high clear sky supersaturations that get more frequent with increasing altitudes and decreasing temperatures in the TTL (e.g. Krämer et al., 2009). While those observations both have received explanations, the analysis presented above suggests that a new mechanism (wave-driven localization) may also contribute.

First, regarding 1), the explanation usually invoked is that the presence of ice crystals damps relative humidity variations towards saturation, by absorbing or releasing water molecules depending on the relative humidity. Using a theoretical parcel model framework, Korolev and Mazin (2003) explain this by the relaxation towards a quasi equilibrium supersaturation state, close to $RH_i = 100\%$. In their stochastic parcel model driven by colored noise temperature variability, Kärcher et al. (2014) also found a damping of the initial mean supersaturation by the ice crystals and a stabilization of the relative humidity with small fluctuations around 100%. In both studies, large excursions away from $RH_i = 100\%$ were prevented by the stabilizing effect brought by the presence of the ice crystals.

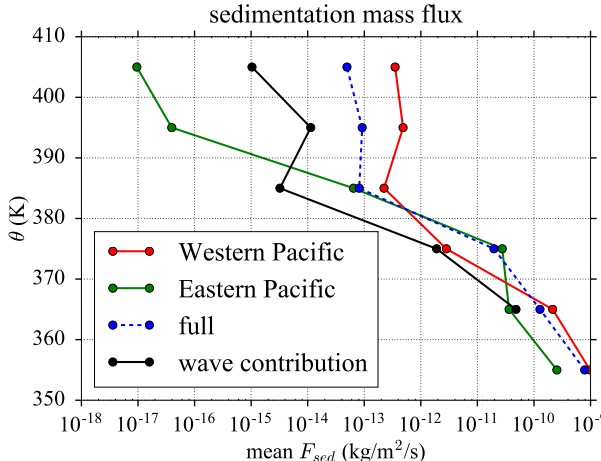

**Figure 7.** Sedimentation water flux $F_{\text{sed}}$ estimated from ATTREX observations in the eastern Pacific 2013 observations, western Pacific and for the whole campaign as a function of potential temperature. The wave stability impact is illustrated for the whole campaign by the black curve, which corresponds to the wave contribution to the flux $\left| F_{\text{sed}} \left( \frac{\partial \theta'}{\partial z} \middle/ \frac{\partial \theta}{\partial z} \right) \right|$. In the upper part of the TTL the wave stability impact reduces the sedimentation flux by about 10% of the total flux.

*In the simulations presented above (Fig. 6), the average relative humidity in clouds is also close to 100%, but for a different reason. Indeed, contrary to the traditional explanations, the feedback of the cloud particles on the ambient relative humidity is **not** included in our simulations. Hence, ice crystals cannot regulate the vapor field towards $RH_i \simeq 100\%$. What happens on the contrary is that only the ice crystals initially located near $RH_i \simeq 100\%$ remain and are constrained to follow the saturated regions due to the wave-driven localization.*

Regarding point 2), it is thought that the increased occurrence of high supersaturations encountered in clear sky at low temperatures (Krämer et al., 2009) is due to the increase with decreasing temperature of the supersaturation threshold required for nucleation. These higher clear-sky supersaturations are often observed in the very cold TTL and the absence of clouds in such conditions limits the efficiency of dehydration (Rollins et al., 2016). In our simulations (see e.g. Fig. 6), we also note that the amplitude of the $RH_i$ oscillations increases with altitude (whereas the temperature amplitude remains constant). The reason for these higher supersaturations at low temperatures in our simulations is the non-linearity of the Clausius-Clapeyron relation with respect to temperature, combined with constant gravity wave temperature amplitude through the TTL. This may be seen from Eq. 18 (which is derived from Clausius-Clapeyron equation): wave-induced $RH_i$ oscillations have their amplitude $\Delta RH_i$ equal to:

$$\Delta RH_i = RH_{i_c}(\bar{Z}) \, \beta_G(\bar{T}) \, Z_{\text{wave}} \tag{35}$$

In our simulations, $\beta_G(\bar{T})$ is the only factor that does depend on altitude (through the background state temperature gradient $\bar{T}(z)$). Given that $\beta_G(\bar{T})$ increases with decreasing temperature (see Eq. 18) and the other factors are constant, it follows that high supersaturation (and subsaturation) are more common (and larger) at low temperatures.

Of course, our simplifications and the above considerations are not strictly applicable to the Earth cirrus clouds, which behave in a more complicated manner. For instance, in the TTL, clouds characterized by $RH_i \simeq 100\%$ generally have large numbers of ice crystals (Jensen et al., 2013; Rollins et al., 2016), which is consistent with the vapor-quenching explanation. Furthermore, even for low ice crystal number TTL cirrus, the assumption made here of negligible water consumption is unrealistic. Damping of the super and subsaturation by the moving ice crystals will tend to broaden the equilibrium regions of $RH_i \simeq 100\%$. Hence, we do not claim that the quenching does not occur and lead to $RH_i \simeq 100\%$, but it is worthwhile to note that another mechanism can lead to the same relation. Our idealized examples may thus point to a process overlooked up to now.

## 4.2 Wave advection versus variability in sedimentation speed

The relevance of wave-driven localization to cirrus clouds evolution and dehydration efficiency depends on their microphysical properties: size distribution and crystal shape. Those points are explained below.

First, regarding the size distribution, it is important to note that our considerations are relevant for crystals whose sedimentation speed $v_{\text{sed}}$ is close to $c_{\phi_z}$, which limits the usefulness to rather small ice crystals (a few tens of microns for the maximum dimension). However, the sedimentation flux $F_{\text{sed}}$ depends on the fourth or fifth-order moment of the size distribution (depending on the regime in which the crystals fall, Kärcher et al., 2014) and is hence more tied to the large ice crystals. The wave impact might hence be negligible if the crystals that dominate $F_{\text{sed}}$ are large compared to those that are affected by the wave-driven localization. To investigate whether this is the case, we show in Fig. 8 the mean size distribution of the number, mass and mass flux of ice crystals from the ATTREX FCDP and 2DS cloud probes during the 2014 field campaign. The mass and sedimentation flux are computed assuming spherical particles for the FCDP bins which ; for the 2DS, the maximum dimension, area and mass measured are directly used. The sedimentation speed needed for the mass-flux computation follows the work of Heymsfield and Westbrook (2010) with the measured maximum dimension, surface area and mass used as inputs to the formulas. Small ice crystals with sizes of a few tens of microns diameter or less dominate the number distribution, with a peak around $10\,\mu$m corresponding to fall speeds around a few mm/s. However, larger ice crystals become more important when the mass distribution is considered, and even more for the mass flux distribution. Nevertheless, the observations suggest that in the TTL, at levels higher than $\theta = 360$ K, the mass flux is still dominated by crystals with maximum dimension smaller than $50\,\mu$m. In that case, the wave-sedimentation interaction described above will have an influence on the dehydration efficiency, since $v_{\text{sed}}$ is not much larger than $|c_{\phi_z}|$.

Second, regarding crystal shape, we have been assuming so far that the ice crystals all have spherical shapes. Although this is mostly the case in observations of crystals with largest dimension below 65 $\mu$m, larger crystals are clearly aspherical (Lawson et al., 2008), which can strongly diminish their sedimentation speed (Jensen et al., 2008): for instance, the sedimentation speed for hexagonal plates might be diminished by 40% relative to the spherical case (Jensen et al., 2008; Westbrook, 2008). This is especially important since those largest, aspherical ice crystals dominate $F_{\text{sed}}$. Hence, the sedimentation flux may be more

affected by the microphysical characteristics of the crystals (their shape,...) than by the wave advection effect. Nevertheless, ATTREX observations do suggest that this effect is present. This might be related to the fact that, at high altitude, a large portion of the mass flux $F_{\text{sed}}$ is associated with crystals below 65 $\mu$m maximum dimension (Fig. 7).

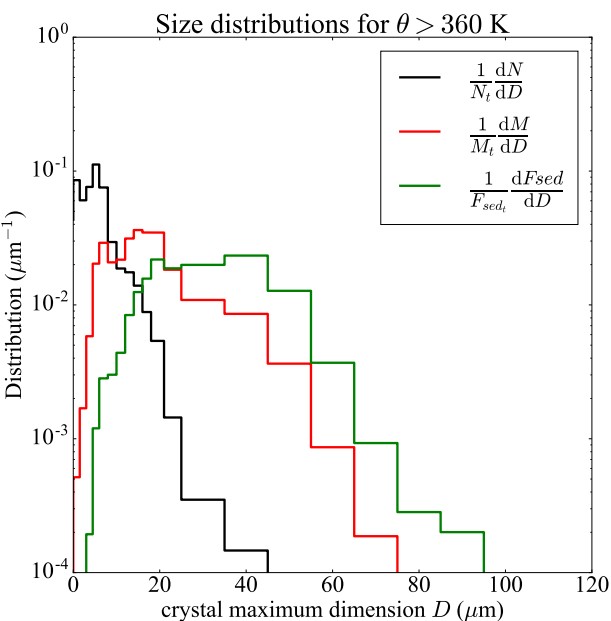

**Figure 8.** Average ice crystal size distribution within cirrus clouds during ATTREX 2014 flights, above $\theta = 360$ K. In black, number distribution; in red, mass distribution; in green, mass flux distribution. The distributions shown are composites of 2DS and FCDP measurements. The size considered is the maximum dimension, the diameter for spherical particles. It should be mentioned that the size retrieved by the FCDP is not strictly exact because the retrieval of size distribution from scattered light assumes spherical particles. The sedimentation speed needed for the mass flux computation are computed using the formula of Heymsfield and Westbrook (2010).

### 4.3 Representation in models

Our calculations emphasize the importance of an accurate representation of the waves for cirrus modeling. In particular, the simulations presented in Sect. 3.1 show that in the presence of the wave, *ice crystals can survive in regions where the background environment is subsaturated*. Climate models lack the vertical resolution to represent fine vertical scale equatorial waves and such processes are in a large part absent. For operational weather prediction models, the situation is less critical, but Podglajen et al. (2014) found that large disagreements could arise between the winds represented in those products and the observed ones.

The role of equatorial and gravity waves has long been acknowledged in Lagrangian cirrus cloud models, which generally include a parameterization of wave fluctuations (e.g. Jensen and Pfister, 2004; Kärcher and Haag, 2004). Most of the time,

however, only temperature fluctuations are accounted for, and the wave advection is ignored. The studies by Jensen et al. (2008) and by Großand Müller (2007) are exceptions, and the particle-following approach remains rarely used with most Lagrangian models only following air parcels. In purely air-parcel following models (e.g. Fueglistaler and Baker, 2006; Spichtinger and Cziczo, 2010), a removal time is prescribed for the falling ice and the wave-driven localization is entirely absent. Column models in isentropic coordinates, such as the one of Jensen and Pfister (2004) and Ueyama et al. (2015) partly include this effect, but they neglect horizontal wind vertical shear which can modify the wave impact on the crystals'motion (see Sect. 3.1). By diminishing the average downward speed of the ice crystals, wave-advection first limits the dehydration of the TTL, and then increases its average cloudiness by keeping surviving crystals within it. Neglecting that effect could add significant uncertainties in our understanding of the water budget of the TTL and lower stratosphere.

## 5   Conclusions

We have investigated analytically the impact of a monochromatic gravity wave on the motion of ice crystals. For an upward propagating GW packet, assuming no water vapor release or depletion by the crystals, an interesting *wave-driven localization* effect is found, in which some of the ice crystals remain confined in a specific wave phase. This wave phase is characterized by positive vertical winds, which slow down the fall of the crystals.

The existence of the *wave-driven localization* is confirmed by idealized numerical simulations of ice crystals growth/sublimation and sedimentation under tropical tropopause conditions. We restricted ourselves to the case of a monochromatic large-scale wave, and did not consider ice nucleation or small-scale gravity waves, which should both be investigated by future work. Despite those limitations, our results provide a plausible and rather simple explanation for the relationship between waves and cirrus clouds observed in the tropical Pacific TTL by Kim et al. (2016). Indeed, in situ observations during ATTREX show that the cirrus are associated with negative temperature anomalies in dry regions (over the tropical eastern Pacific) and with negative vertical temperature gradient anomalies in moister region (over the tropical western Pacific). This observational finding is consistent with both our analytical results and our numerical simulations. Furthermore, from ATTREX observations, the wave-driven localization and wave wind advection might diminish the ice flux leaving the TTL by about $10\%$ relative to the flux estimated ignoring the wave-induced vertical wind variations. This is due to the more frequent occurrence of positive vertical winds where the ice crystals are present.

Two main conclusions can be drawn from our study. First, there is a fundamental difference between air parcels and ice particles: due to wave-driven localization, waves can have an average impact on the motion of ice crystals even if they have none on air parcels'. Second, water vapor quenching by the ice is *not* the only mechanism to consider when examining the distribution of relative humidity in clouds. Ice crystals motion in a variable relative humidity field is also highly relevant and should not be overlooked. Indeed, wave-driven localization guides ice crystals to concentrate where $RH_i \sim 100\%$ without any effect of the ice crystals on the relative humidity.

Although we focused on TTL cirrus, the theory introduced here is general and might be relevant to other types of clouds or aerosols which are largely influenced by waves and the associated wind field. An example is the case of noctilucent clouds

(NLC) in the summer polar mesopause region. Rapp et al. (2002) have investigated the influence of waves on NLC and demonstrated that the interplay between sedimentation, transport by the wave vertical wind, and growth could lead to the NLC layer following the motion of the cold phase of the wave. However, if those authors also used particle trajectory calculations and emphasized the role of vertical wind in competition with sedimentation, they neglected the impact of the horizontal wind which

can significantly modify the wave impact on sedimentation (see Sect. 3.1 and appendix C). Another example of microphysical process strongly affected by waves is the case of Polar Stratospheric clouds, which are affected by mountain waves over the Antarctic peninsula or the Scandinavian mountains (e.g. Carslaw et al., 1999) and could show similar relations as presented here.

Besides the atmosphere, many other media, such as oceans and lakes, are perturbed by internal gravity waves. Those waves

contribute to particle transport through the classical Stokes drift, mixing due to wave breaking and also through the resuspension of sediments due to the stress exerted by the induced flow on the bottom (Cacchione et al., 2002; Butman et al., 2006). Although it involves different physics, a parallel can be made between this last process and the irreversible wave-induced motion we describe: in both cases it is the combination of the wave presence and the particles' motion in relation to the flow that lead to irreversible transport by the wave. The effect of the waves on the sedimentation of particles (described in Sect. 2.1, before

considering growth and decay of ice crystals) might also directly play a role in sediment transport in oceans and lakes or particle transport in the atmosphere of any planet. Indeed, the theoretical derivation in Sect. 2.1 applies to any stratified fluids perturbed by internal waves and containing particles in suspension (that is, almost any stratified fluid).

Specifically for the atmosphere, a number of previous works have stressed the impact of waves on UTLS clouds. Here, we have unraveled yet another effect, which possibly provides an explanation for recently observed relations between waves and

20 cirrus. Our results hence call for more quantitative observations of waves and cirrus clouds, in order to more precisely nail down the wave impact on clouds and improve its representation in models. Quasi-Lagrangian cloud and wave observations coupled with particle-following model simulations would be especially convenient to investigate this effect.

*Data availability.* The ATTREX aircraft data used in this paper can be retrieved from NASA Earth Science Project Office (ESPO) at https://espoarchive.nasa.gov/archive/browse/attrex.

## Appendix A: Note on wave stability

To avoid strongly overestimating the effect of wave driven vertical transport on sedimentation, it is important to recall that the wave amplitude is limited by a stability requirement. Shear and convective breaking of monochromatic gravity waves have been treated in a number of studies (e.g. Lindzen, 1981), and we adapt those considerations to our notations, to highlight where our two wave parameters ($c_{\phi_z}$ and $W$) intervene.

We take the criterion that the Richardson number $Ri$ must be larger than a critical value $Ri_c$. Miles-Howard stability criterion suggests $Ri_c \simeq \frac{1}{4}$, and a more conservative choice would be $Ri_c \simeq 1$. For simplicity and consistency with our configuration,

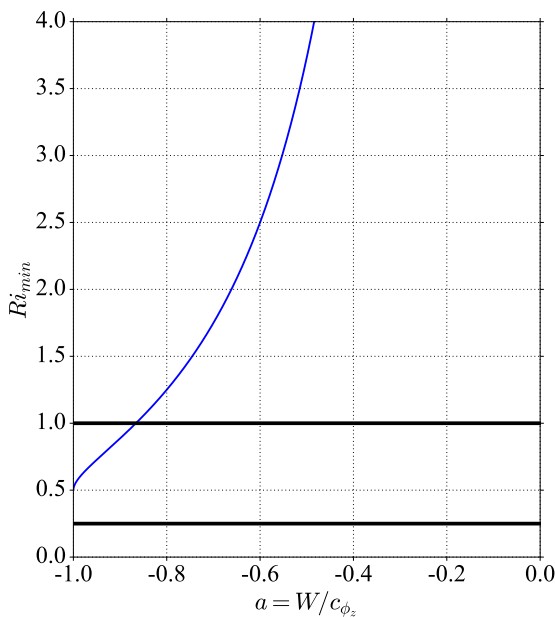

**Figure A1.** Minimum Richardson number over all wave phases, as a function of the amplitude parameter $\frac{W}{c_{\phi_z}}$.

we neglect the background shear; then the Richardson number induced by the monochromatic wave field is:

$$Ri = \frac{\frac{g}{\theta}\frac{\partial\theta}{\partial z}}{\left(\frac{\partial u}{\partial z}\right)^2} = \frac{\bar{N}^2(1+\frac{W}{c_{\phi_z}}\cos(\Phi))}{\frac{m^4}{k^2}W^2\sin^2(\Phi)} = \left(\frac{c_{\phi_z}}{W}\right)^2\frac{1+\frac{W}{c_{\phi_z}}\cos(\Phi)}{1-\cos^2(\Phi)} = \frac{1}{a^2}\frac{1+a\cos(\Phi)}{1-\cos^2(\Phi)} \tag{A1}$$

with $a = \frac{W}{c_{\phi_z}}$ ($a < 0$ since we are interested in upward propagating waves). Figure A1 represents the minimum Richardson number over all wave phases in our configuration. Examination of the figure or of Eq. A1 shows that for $a \geq -1$, $Ri \geq 0.5$.

5   The condition $Ri > \frac{1}{4}$ is equivalent for the monochromatic upward propagating wave to the condition for convective stability, i.e. $Ri > 0$ or $|a| < 1$. The condition $Ri > 1$ is a bit more restrictive and requires $0 \geq a > \sim -0.82$. We will then restrict the chosen wave amplitude so that the minimum $Ri$ remains above 1, i.e. $|a| < 0.82$. Wave breaking and the generated turbulence probably has an important impact on vertical mixing and vertical transport of ice and water in the TTL (e.g. Podglajen et al., 2017), but this is not considered in our configuration which focuses on propagating waves.

## Appendix B: Hamiltonian structure of the equations

Writing $p = \Psi$ and $q = r^2$, Syst. (20) reads :

$$\begin{cases} \dfrac{dp}{dt} = -Aq - B \\ \dfrac{dq}{dt} = C\sin(p) + D \end{cases} \tag{B1}$$

so that, with $H(p,q) = \frac{A}{2}\left(q + \frac{B}{A}\right)^2 - C\cos(p) + Dp$, we have $\frac{dp}{dt} = -\frac{\partial H}{\partial q}$ and $\frac{dq}{dt} = \frac{\partial H}{\partial p}$. The system is therefore Hamiltonian, and the trajectories in the $p-q$ space are given by the curves of constant $H$.

The existence of periodic orbits is equivalent to the existence of local extrema of the Hamiltonian function. Noting that the existence of fixed points (and extrema of $H$) for $q > 0$ requires that $\omega$ and $m$ are of opposite signs, which implies that $A$ and $C$ are of the same sign. Given its expression, local extrema of $H$ for $q > 0$ can then only be encountered for $q = -B/A$. Hence there are local extrema of $H$ iif there exists $p \in [0; 2\pi]$ for which $\frac{\partial H}{\partial p}(p, -B/A) = C\sin(p) + D$ cancels and changes sign and for which $\frac{\partial^2 H}{\partial p^2}(p, -B/A) = C\cos(p)$ is of the sign of $A$ ($= \frac{\partial^2 H}{\partial q^2}$). It is easy to see that there exists such $p$ iif $\left|\frac{D}{C}\right| < 1$; since $A$ and $C$ are of the same sign, those extrema, which correspond to the elliptic points, are located where $\cos(p) > 0$.

The expression of $H$ also allows to evaluate the location of the separatrix which delimits the region of periodic orbits, when they exist. Noting $p_e$ the phase of the elliptic point and $p_s$ that of the closest saddle point, and assuming $A > 0$, it can be seen from geometric arguments that the coordinates $(p, q)$ of the separatrix are characterized by:

$$H(p, q) = H(p_e, q = 0) \tag{B2}$$

if $H(p_s, q = -B/A) > H(p_e, q = 0)$, otherwise by

$$H(p, q) = H(p_s, q = -B/A) \text{ and } p \geq p_s \tag{B3}$$

The first case arises because $q = r^2$ is physically required to be larger than $0$. The general mathematical requirement for the separatrix (if negative $q$ were possible) is the second case. Given those constraints, it is possible to find numerically the phase $p = \Psi_{sep}$ of the intersections of the separatrix with the $r = r_f$ line, and hence the fraction $\mathcal{F}_p$ of the phase space affected by that.

## Appendix C: Impact of neglecting the horizontal wind induced by the wave

The theoretical analysis presented in the paper accounts for both the horizontal and the vertical wind component due to the wave. Figure C1 is similar to Fig. 6 but also shows the ice crystal's position when the horizontal wind induced by the wave is neglected. In the low $RH_i$ case (left panel), contrasting the crystal positions between the full wind wind and no horizontal wind simulations reveals a tendency of the remaining ice crystals to be present at higher altitude when full wave advection is accounted for. This difference between the full wind wind and no horizontal wind simulations is even more striking for the high

$RH_i$ scenario (right panel). Furthermore, when the horizontal wind is neglected, the proportion of ice crystals surviving after half a wave period is increased by a factor 2 (moist case) to 3 (dry case). This shows the limits of the single-column approach.

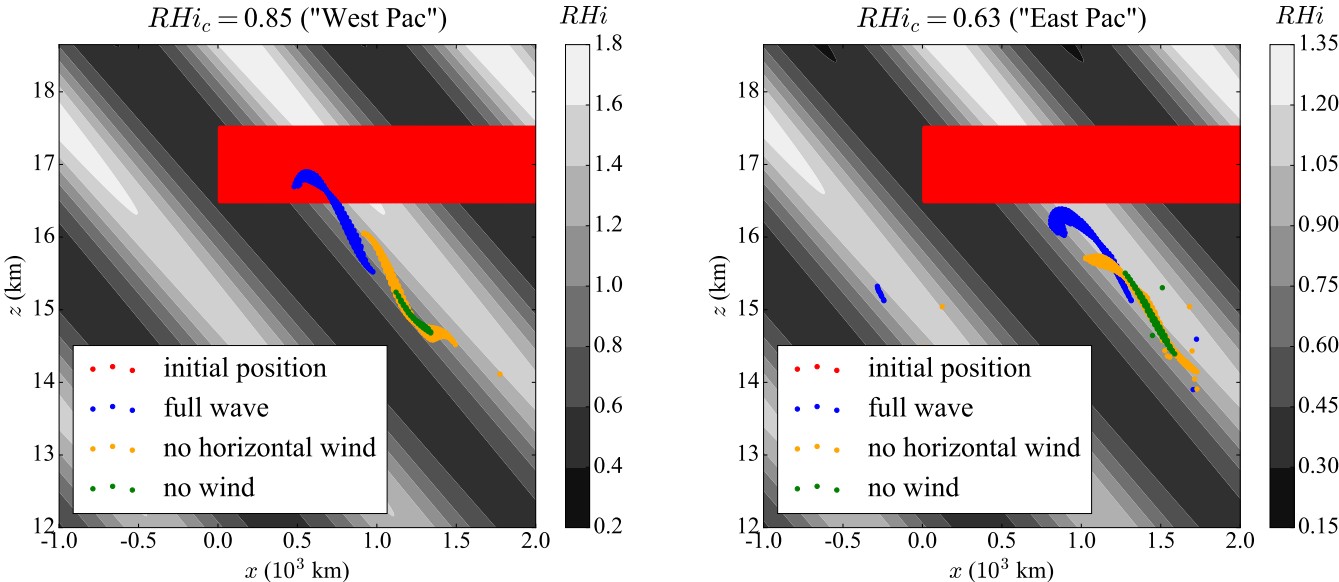

**Figure C1.** Same as Fig. 6, but including ice crystal positions from simulations with the wave vertical wind accounted for but not the horizontal one (orange). As in Fig. 6, the blue dots are the ice crystals' positions for the full simulation (wave advection and temperature fluctuations), and the for the green dots both the horizontal and vertical winds induced by the wave have been neglected. The initial ice crystal radius used in both cases for all crystals is 5 $\mu$m.

It might seem surprising that, once the horizontal wind is neglected, the downward speed of the crystals at the elliptic point $w_c$ is similar whether or not the vertical wind is accounted for (see the orange and green dots in figure C1). This is due to the

5 fact that when the vertical wind is taken into account but not the horizontal wind, the sedimentation speed at the elliptic fixed point is:

$$v_{\text{sed}_{\text{no hor}}} = c_{\phi_z} - W\cos\Psi_f < c_{\phi_z} = v_{\text{sed}_{\text{no wind}}} \tag{C1}$$

so that the crystals have higher fall speeds and larger sizes than when neither the vertical nor the horizontal wind speeds are accounted for.

10 With only the wave vertical wind, the total vertical speed of the crystals $w_c$ is the sum of the vertical wind and the fall speed:

$$w_{c_{\text{no hor}}} = v_{\text{sed}_{\text{no hor}}} + W\cos\Psi_f \tag{C2}$$

while with no wave wind the total vertical speed is just the crystal fall velocity:

$$w_{c_{\text{no wind}}} = v_{\text{sed}_{\text{no wind}}} \tag{C3}$$

Hence, interestingly, taking into account only the wave vertical wind increases the sedimenting ice flux at the elliptic point compared to the full wind and no wind cases, due to the increase of the size of the crystals when the vertical wind component is accounted for. This increase of the sedimenting mass flux occurs even so the positive vertical winds where the ice crystals are present would be expected to diminish that flux. However, one should recall that only the ice crystals confined near the elliptic point are concerned by this effect.

## Appendix D: Clouds and stability during ATTREX

This appendix compares the stability ($\frac{\partial \theta'}{\partial z}$) within and out of clouds in ATTREX observations, using cirrus cloud observations and vertical temperature profiles form the Microwave Temperature Profiler. The results, consistent with those of Kim et al. (2016), complement them using the isentropic approach.

Figure D1 shows the mean lapse rate as a function of potential temperature, within and out of clouds. In the upper TTL (above 380 K or about 16.5 km) over the western Pacific, the potential temperature lapse rate appears systematically lower within clouds compared to out of those, which is consistent with the results of Kim et al. (2016) who found anomalous negative lapse rate of wave temperature anomalies $dT'/dz$ within clouds. This difference between cloudy and cloud free air is robust despite the limited number of independent events, and seen in different flights, in particular it is larger than the vertical change of stability in the upper TTL, so that it is not due to clouds being more prevalent in the lower part of the bins where stability is weaker.

This relationship between clouds and stability is further analyzed in Fig. D2, which presents the observed probability distribution of $\frac{\partial \theta'}{\partial z}$ for cloudy air versus clear air, in the eastern and western Pacific and in different potential temperature ranges. In the eastern Pacific and in the western Pacific above 380 K, Figure D2 shows a peak of the cloudy-air PDF in negative $\frac{\partial \theta'}{\partial z}$. The peak is shifted towards lower values of $\frac{\partial \theta'}{\partial z}$ in the western Pacific above $\theta = 380~K$ than in the eastern Pacific, consistent with the results of Kim et al. (2016). Indeed, Kim et al. (2016) found that, in the western Pacific above 15 km, most clouds were characterized by negative $dT'/dz$ whereas in the eastern Pacific clouds were seen both in positive and negative $dT'/dz$. *The picture provided by Fig. D2 is also consistent with the sensitivity to background moisture in the idealized and more realistic simulations results shown in Fig. 2 and 6: in the dry eastern Pacific, the ice crystals are more common in the minimum temperature, $dT'/dz \simeq 0$ phase of the wave. In the moister western Pacific at high altitude, the ice crystals are focused in the $dT'/dz < 0$ phase of the wave.* In both cases, that phase is the phase where $RH_i \simeq 100\%$.

*Competing interests.* The authors declare that they have no conflict of interest.

*Acknowledgements.* The authors thank the teams involved in the development and exploitation of the MMS, MTP, FCDP and 2DS instruments during the ATTREX campaign. AP thanks Martina Krämer, Bernard Legras and Claudia Stubenrauch for their comments on this

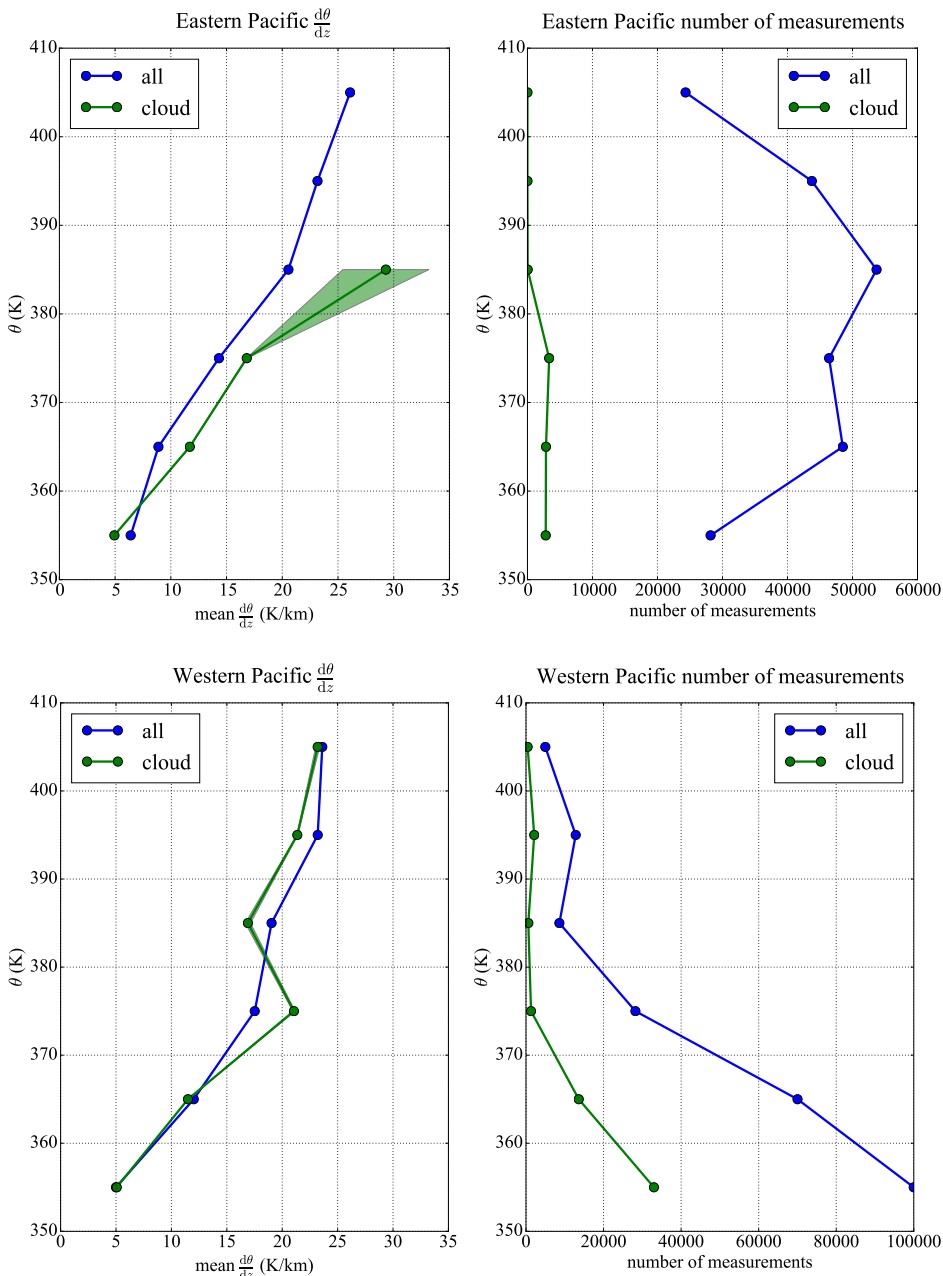

**Figure D1.** (Left panels) Average potential temperature vertical gradient $\frac{\mathrm{d}\theta}{\mathrm{d}z}$ in cloudy and cloud-free air in the eastern (top) and western Pacific (bottom). The right panels show the total and in cloud number of measurements as a function of altitude. On the left panel, the $1 - \sigma$ uncertainties are represented by shadings, but the amount of measurements are sufficient to make them indiscernible. This is nevertheless no statistical proof since the data come only from a few flights and are correlated.

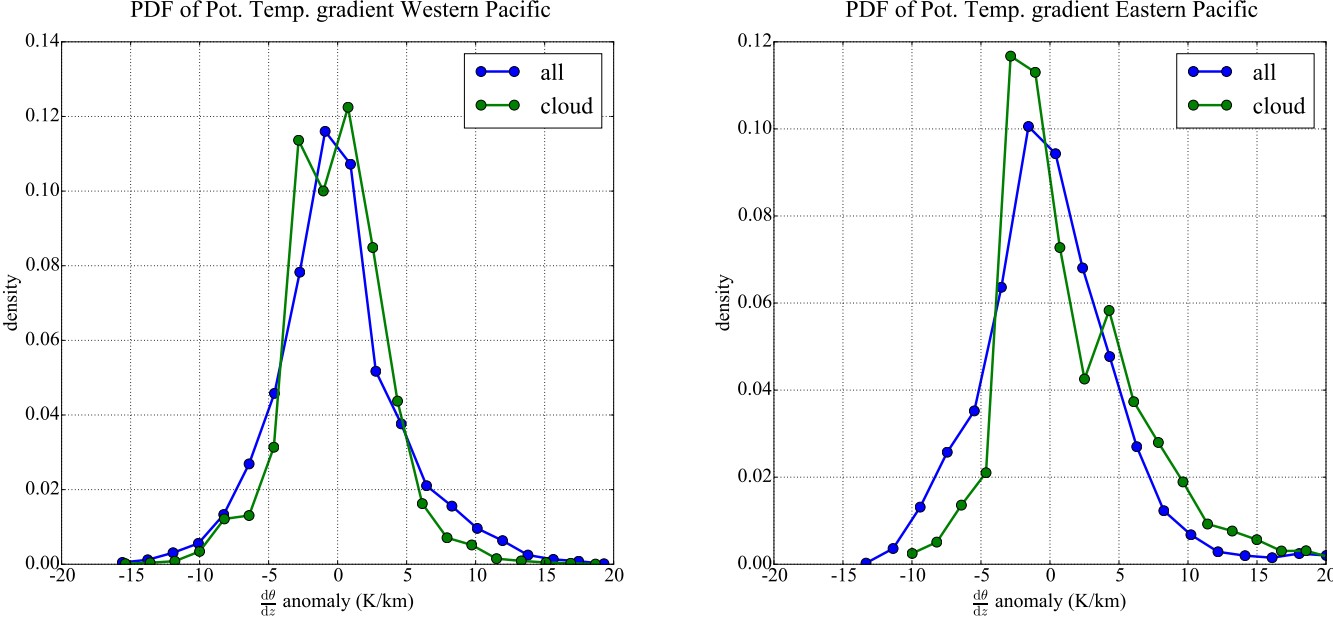

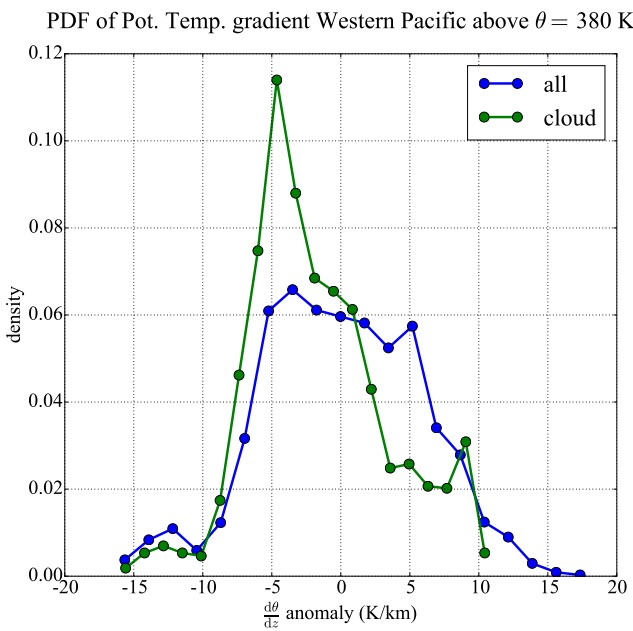

**Figure D2.** Anomalies of stability $\left(\frac{\partial\theta}{\partial z}\right)$ distributions out of and within clouds from ATTREX observations in the (left) western Pacific, 2014 and (middle) eastern Pacific, 2013. The right panel also corresponds to the western Pacific but only data above the potential temperature level 380 K.

work. We sincerely thank the two anonymous reviewers for their helpful comments and suggestions on the manuscript. AP, RP, and AH acknowledge support from the french ANR project StraDyVariUS (Stratospheric Dynamic and Variability, ANR-13-BS06-0011-01)

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
