# Peer review of "Impact of gravity waves on the motion and distribution of atmospheric ice particles"

_Atmospheric Chemistry and Physics, 2017_

## Referee Comment (RC1) · Anonymous Referee #1 · 30 Nov 2017

**General remarks**

This investigation points to an effect in cirrus clouds that has so far been largely over-looked, called wave-driven localization. It means that by the combined action of waves, crystal sedimentation and crystal growth/sublimation it can happen that crystals collect in a region where the relative humidity wrt ice is about 100%. The important consequence of this is that the lifetime of those crystals is longer than without the waves since the crystals cannot fall away from that "elliptic fixed point"; this in turn might reduce dehydration and increase the occurrence of thin cirrus in the TTL.

This is an interesting paper, with a high quality of its mathematical derivations and numerical applications. It is worth publication in ACP.

[Figure]

That said, I must admit that I am not convinced of the relevance of the localization effect for the atmosphere. This remains to be demonstrated. There are two major reasons for my scepticism:

**Major points**

1) There are a number of simplifications, necessarily in the analytical model, and in the numerical model. For many of these there may be good reasons or they are harmless (spherical crystals). But there are two simplifications that may be critical.

One is the assumption that crystals are already there at the initialisation of the model. On page 18 the authors state " What happens ... is that only the ice crystals INITIALLY located near $RH_i \approx 100\%$ remain ...". As ice nucleation usually needs high supersaturation, I wonder whether there are ever ice crystals initially at 100%.

The other critical assumption is that of a negligible feedback of crystal growth/sublimation on RHi. As the authors say, high crystal concentrations are common in TTL cirrus, so that assumption might be unrealistic. To my view, it is an assumption that might be necessary to develop the theory and the arguments, but later it could be relaxed. It should be relatively easy to run the numerical model with water-ice feedback. The question then is whether the localization effect is still present when the feedback is switched on.

However, actually there are cirrus clouds in the TTL that have extremely small crystal number concentration. These are the "Ultrathin Tropical Tropopause Clouds (UTTCs)" (Peter et al., 2003). Luo et al.(2003) have proposed a mechanism that leads to a stabilisation of such clouds. I suggest that the authors mention the UTTCs and the corresponding mechanism, although it works without waves. Also Spichtinger and Krämer (2013) proposed a mechanism that would produce clouds with low crystal numer densities; their mechanisms works with short waves where the wave "down phase" essentially terminates the ongoing nucleation process. I think this work should also be mentioned and the difference between the proposed mechanisms briefly discussed.

2) On page 20 (last lines) the authors make the point that the localization is an important effect and that its disregard in global models with their course vertical resolution and in weather models leads to "significant uncertainties". To my view, this is too cheap a statement. The statement may be ok if it had been written in conditional tense and without the "significant". Otherwise, it must be shown what the bad consequences of its negligence are on dehydration, radiation, water vapor transport into the tropical stratosphere, etc.

**1   Minor points**

1) Page 2, Line 21/22: As the wave phase is a purely mathematical object, I suppose that it can only affect ice crystals indirectly. An influence can only be exerted by material (physical) properties of the crystals environment, as T or RH. Does your statement imply that such properties are uniquely related to the wave phase?

2) P. 3, L. 2: Is it possible at all that RH=const in a wavy environment? Perhaps in this special case you better speak of "solid particles that fall but that don't grow or sublimate" instead of ice.

**2   Tiny points**

1) Page 1, Line 22/23: Isn't the wind identical to the movement of air parcels?

2) P. 2, L. 2: 190 K is not a range.

3) P. 2, L. 3: insert "of the" before "atmosphere".

4) P. 2, L. 15: write "to and fro" instead of "to and from".

5) P. 2, L. 20/21: "the falling particles fall in the same direction as the wave phase" implies that the phase falls. Better write "the falling particles fall in the direction of wave propagation".

6) P. 2, L. 32: which system?

7) P. 5, L. 3,4: "green" should be "red".

8) P. 5, L. 26: Although the notion "perfect gas" exists (a further simplification of an ideal gas), the gas constant should be termed "gas constant" or "specific gas constant for water vapor". There is nothing in the calculation presented that needs the assumptions of a "perfect gas".

9) P. 7, L. 5: please write "crystal number concentrations".

10) P. 7, L. 27: The "next section" is 2.2.3, not 2.2.1.

11) Eq. 18: I am puzzled by the terms $RH_{i_c}(\overline{Z})$. Before $RH_{i_c}$ was introduced as a constant. Why is it now a function of $\overline{Z}$? Please mention also the meaning of the terms in the brackets (probably Clausius-Clapeyron and pressure change?).

12) P. 8, L. 17: As $RH_{i_c}$ was never specified, is the fix point possible for the whole range of possible values? Is it tacitly to be understood that $RH_{i_c}$ is close to or above ice saturation since there are ice crystals?

13) P. 9, L. 3: Check sentence!

14) P. 9, L. 13: Which of the amplitudes?

15) Fig. 2: Is it possible to indicate the direction of the motion in phase space?

16) P. 11, L. 11-12: On first reading, it was not clear to me what exactly is the difference between the "cold" phase in the eastern Pacific and the "cooling" phase in the western Pacific. Only the later reference to figure 2 clarifies that. I suggest to refer earlier to the

figure to illustrate the distinction.

17) Figure 4 does not show green points, contrary to what the caption says.

18) P. 14, L. 17: blue is a color as well!

19) P. 16, last par: change "equality" to "equation".

20) Figure 6: blue and black are hard to distinguish.

21) P. 18, L. 15: The sentence is a bit strange. In clear sky there are no cirrus clouds. How can then their dehydration efficiency be constrained?

22) P. 19, L. 6: "order" should be "power".

23) P. 19, L 30.: "disagreement" between what?

References

Luo, B.P., Peter, Th., Wernli, H., Fueglistaler, S., Wirth, M., Kiemle, C., Flentje, H., Yushkov, V.A., Khattatov, V., Rudakov, V., Thomas, A., Borrmann, S., Toci, G., Mazzinghi, P., Beuermann, J., Schiller, C., Cairo, F., Di Don-Francesco, G., Adriani, A., Volk, C.M., Ström, J., Noone, K., Mitev, V., MacKenzie, R. A., Carslaw, K. S., Trautmann, T., Santacesaria, V., and Stefanutti, L.: Ultrathin Tropical Tropopause Clouds (UTTCs): II. Stabilization mechanisms, Atmos. Chem. Phys., 3, 1093-1100, https://doi.org/10.5194/acp-3-1093-2003, 2003.

Peter, Th., Luo, B.P., Wernli, H., et al.: Ultrathin Tropical Tropopause Clouds (UTTCs): I. Cloud Morphology and Occurrence, Atmos. Chem. Phys., 3, 1083–1091, 2003.

Spichtinger, P., Krämer, M.: Tropical tropopause ice clouds: a dynamic approach to the mystery of low crystal numbers. Atmos. Chem. Phys., 13, 9801–9818, 2013.

---

## Referee Comment (RC2) · Anonymous Referee #2 · 12 Feb 2018

Review of
**Impact of gravity waves on the motion and distribution of atmospheric ice particles**
by Aurélian Podglajen et al.

**General comment:**
In this study a new mechanism for influencing ice crystal distributions in the tropical tropopause layer (TTL) is proposed, a so-called "wave-driven localization" of ice crystals. The mechanism is investigated with an idealized (toy) model, reproducing the main features, with numerical simulations and comparisons with observations from a recent campaign (ATTREX).

Overall, this is a very interesting study including dynamical effects from waves on different scales on ice microphysics and ice clouds in the TTL; thus this is an adequate and meaningful contribution to ACP. However, there are some issues which should be clarified before the manuscript can be accepted for publication. Therefore I recommend major revisions for the manuscript.

In the following I will explain my concerns in detail.

**Major points**

1. Description and analysis of the simplified ODE system:
   Generally, it is a very meaningful approach to formulate a simple model for representing the important processes and to use this model for a rigorous analysis; this is also a very interesting and important result of this study.

   However, this part of the manuscript should be revised and partly rewritten, since it is very difficult to follow the line of arguments. This is mostly due to the very irritating notation, which is changed in the section several times. For instance, new coefficients as $\alpha_G$ are introduced but only partly used. Sometimes the text refers to "the first" or "the second" equation, but it is not really clear, which equations are meant. In fact, the restriction for the relative humidity to be equal to 100% does not follow from the "second" equation (19) but from the requirement for the equilibrium point, that the derivatives must be zero and thus the radius can only be constant, if the cloud is in thermodynamic equilibrium.

   Beside the confusing (but nevertheless correct) description of the model system and the linearization, there is a major problem for the correct analysis of the nonlinear system. The qualitative behaviour of the equilibrium point in the linearisation can only be transferred to the original nonlinear system, if the eigenvalues have non-zero real part (hyperbolic points). Thus, for the saddle point the argumentation is correct. For points with eigenvalues of zero real part (non-hyperbolic points), the quality of a centre point (in the linearization) cannot easily be transferred to the non-linear system (see, e.g., Verhulst, 1996 or Hirsch et al., 2013).

   I would suggest (also in terms of simplification of the notation) to rewrite the system using new variables $x = \Psi$, $y = r^2$ and constants $a, b, c, d$:

   $$\dot{x} = -c - dy \tag{1}$$
   $$\dot{y} = -a \sin x + b \tag{2}$$

   This abstract formulation helps to see the formal structure of the equations. In fact, it can be seen easily that the system is Hamiltonian with a Hamilton function as follows (transformation $q = x$, $p = y$):

   $$H(p, q) = -cp - \frac{d}{2}p^2 - bq - a \cos q \tag{3}$$

Using the Hamilton function, the stability of the elliptic point as well as the existence of the periodic solutions can be determined easily. In addition, the Hamilton function might be used for the calculation of trajectories, since solutions are given by $H(p,q) = $ const., and maybe also for determining the domain of attraction around the elliptic point. This might be interesting in the sense, how many ice particles are really influenced by the mechanism, or better, how close the particles must be to the elliptic point to be affected.

Finally, the representation of the solutions and their stability points in figure 2 is quite difficult to understand; the behaviour of the trajectories is not completely clear. In fact, it seems that some trajectories disappear at $r = 0$, which is probably meaningful (evaporation). However, it is not clear what happens in several parts of the phase space, e.g. in the case $RHi_c = 0.85$ in the range $\frac{\pi}{2} \leq \Psi \leq \pi$ or in the range $\frac{5}{2}\pi \leq \Psi \leq 3\pi$. Please reproduce the figure with some zooms around the equilibrium points and less trajectories for explaining the schematic behaviour of the solutions (phase portrait). Maybe this representation can be connected with the important and illuminative physical interpretation in section 2.2.4.

2. Neglecting water vapour depletion by ice crystals:

For the formulation of the model equations (11) and also the simplified model (eq. 19), the depletion of water vapour by crystal growth is neglected. I can understand that for the analysis of the model this is a convenient simplification. However, it should be estimated how large the effect on the background fields as well as the solutions really is. This should be done analytically and/or using numerical simulations. Probably, the effect is really small and the assumption is meaningful but this must be shown.

The whole study treats ice crystals, which are already there, i.e. the formation of ice crystals is not taken into account. However, in principle ice crystals are formed in the low temperature regime of TTL at high supersaturations ($RHi \sim 130 - 170\%$, depending on the formation mechanism). Thus, the assumption of ice crystals in a region at thermodynamic equilibrium seems to be quite strong. For me two different scenarios might be possible, if we start with ice nucleation:

(a) If only a few ice crystal form, they are not able to deplete enough water vapour for reaching equilibrium and thus the described mechanism does not work, until the ice crystals have grown to larger sizes and have fallen out into a region with relative humidity close to ice saturation. It is not clear if for large ice particles (radius close to $100\,\mu\mathrm{m}$) the described mechanism will be efficient. Please, comment on this.

(b) If many ice crystals are formed, they will deplete the water vapour without growing to larger sizes (because they are many) until the system reaches equilibrium. Then the described mechanism can play a role. In this scenario, please describe, how large the effect of wave-driven localization is in comparison to quenching of water vapour.

**Minor points:**

1. Figure 1: Aspect ratio of the phenomenon
In the example of figure 1, the vertical extension is of order $O(3\,\mathrm{km})$ whereas the horizontal extension is of order $O(10^3\,\mathrm{km})$; thus the aspect ratio is very small, please indicate this in the text and also in the figure caption.

2. Page 4, lines 7-15 and following next page:
It seems that the effect of wave-driven localization is mainly effective for waves with quite low frequencies (Kelvin waves). Please comment this in the text.

3. Constraining the value of deposition coefficient:
   Actually, Skrotzki et al. (2013) does give a recommendation for a value of the deposition coefficient, based on a collection laboratory experiments, model simulations and a synthesis of both, i.e. $\overline{\alpha}_d = 0.7$ and $0.2 \leq \alpha_d \leq 1$. Thus, the used value of $\alpha_d \sim 0.5$ is in the recommended range. Please reformulate the text accordingly.

4. Expression for the saturation mixing ratio:
   The correct (but still approximate) formula for the saturation mixing ratio is $q_{\text{sat}} = \epsilon \frac{e_{\text{sat}}(T)}{P}$ with the ratio of molar masses of water and air, respectively, $\epsilon = \frac{M_v}{M_a}$.

5. Figure 4 and text:
   In this figure the time evolution of the particles' position is shown. It would be nice to quantify how many particles from the initial distribution at 0.0 days really survive in a position close to the elliptic point. A similar statistics would be interesting for the simulations in figure 5 and figure B1 in the appendix.

6. Page 15, line 15 and equation (15): Slow down of ice crystal sedimentation
   It is stated here that the sedimentation is reduced significantly by wave advection. Can you quantify this statement, i.e. by which fraction is the sedimentation reduced for distinct conditions?

7. Validity of several approximations
   For the formulation of the model equations some approximations are made without much information about the validity of the approximation, e.g. the assumption of spherical particles (Stokes' flow for sedimentation, eq. 17) or the linearisation of the saturation vapour pressure (eq. 18). Please indicate (at least in the appendix) the validity of these approximations quantitatively. On the other hand, the full growth factor for ice crystals is used, including kinetic and ventilation corrections and latent heat release. Since the model is used in a very small part of the phase space (radius $5\,\mu\text{m} \leq r \leq 100\,\mu\text{m}$, very cold temperatures in the TTL) not all corrections are really meaningful or necessary. Thus, there is a kind of discrepancy between approximations on one hand and very accurate treatment of processes on the other hand. Please resolve this discrepancy in a meaningful way.

**References**

Verhulst, F., 1996: Nonlinear Differential Equations and Dynamical Systems. Second edition, Springer, Heidelberg.

Hirsch, M., S. Smale, R. Devaney, 2013: Differential Equations, Dynamical Systems, and an Introduction to Chaos. Academic Press (Elsevier), Amsterdam.

---

## Author Comment (AC1) · 9 Apr 2018

**Impact of gravity waves on the motion and distribution of atmospheric ice particles: reply to reviewer 1**

April 9, 2018

We would like to thank the reviewer for his/her constructive comments on our manuscript, especially for the suggestion of missing relevant references. Please find below our point-by-point reply.

1. ***Reviewer*** — This investigation points to an effect in cirrus clouds that has so far been largely over- looked, called wave-driven localization. It means that by the combined action of waves, crystal sedimentation and crystal growth/sublimation it can happen that crystals collect in a region where the relative humidity wrt ice is about 100quence of this is that the lifetime of those crystals is longer than without the waves since the crystals cannot fall away from that "elliptic fixed point"; this in turn might reduce dehydration and increase the occurrence of thin cirrus in the TTL. This is an interesting paper, with a high quality of its mathematical derivations and numerical applications. It is worth publication in ACP.

   That said, I must admit that I am not convinced of the relevance of the localization effect for the atmosphere. This remains to be demonstrated. There are two major reasons for my scepticism:

   1) There are a number of simplifications, necessarily in the analytical model, and in the numerical model. For many of these there may be good reasons or they are harmless (spherical crystals). But there are two simplifications that may be critical. One is the assumption that crystals are already there at the initialisation of the model. On page 18 the authors state " What happens ... is that only the ice crystals INITIALLY located near $RH_i \simeq 100\%$ remain ...". As ice nucleation usually needs high supersatu- ration, I wonder whether there are ever ice crystals initially at $100\%$.

   ***Authors*** — Although the wave-driven localization at $100\%$ relative humidity depends on ice crystals being already present there, our use of "initially" in the context of the article does not necessarily refer to the nucleation time of ice particles. We agree with the reviewer that supersaturation is needed to nucleate ice crystals in the TTL, but as the sedimentation starts, ice crystals will likely encounter $RH_i \simeq 100\%$. At this time, the crystal size, the wave characteristics, and the background relative humidity will be critical to determine whether these ice crystals will be sensitive to the wave-localization effect. One could imagine several mechanisms, such as small-scale gravity waves locally increasing the

$RH_i$, to explain the initial formation of ice crystals that subsequently sediment (see also response to reviewer 2). However, including this in our set-up would require a number of additional assumptions that are better left for future investigations. We now emphasize explicitly in the text the "ad hoc" initialisation.

2. **Reviewer** — The other critical assumption is that of a negligible feedback of crystal growth/sublimation on RHi. As the authors say, high crystal concentrations are common in TTL cirrus, so that assumption might be unrealistic. To my view, it is an assumption that might be necessary to develop the theory and the arguments, but later it could be relaxed. It should be relatively easy to run the numerical model with water-ice feedback. The question then is whether the localization effect is still present when the feedback is switched on. However, actually there are cirrus clouds in the TTL that have extremely small crystal number concentration. These are the "Ultrathin Tropical Tropopause Clouds (UTTCs)" (Peter et al., 2003). Luo et al.(2003) have proposed a mechanism that leads to a stabilisation of such clouds. I suggest that the authors mention the UTTCs and the corresponding mechanism, although it works without waves. Also Spichtinger and Kraemer (2013) proposed a mechanism that would produce clouds with low crystal numer densities; their mechanisms works with short waves where the wave "down phase" essentially terminates the ongoing nucleation process. I think this work should also be mentioned and the difference between the proposed mechanisms briefly discussed.

   **Authors** — It is true that our study is more relevant to low ice-crystal number clouds since we have on purpose omitted the feedback of ice crystals on water vapor. This idealized set-up notably enables us to highlight the role of the wave-driven localization effect, which is able to maintain clouds at $RH_i \simeq 100\%$ on its own. When referring to very thin, low ice-crystal number cirrus such as those observed by Jensen et al. (2013, 2017), we were actually already considering UTCCs without using the name. We now explicitly mention the name "UTCCs" and reference Peter et al., 2003 in the revised version paper. We had actually missed the very relevant Luo et al.(2003) reference, which is now discussed (in Sect. 2.2.3). However, the work of Spichtinger and Kraemer (2013) deals with the influence of gravity waves on ice nucleation, a very different problem from that addressed in our work. We now mention their study in the introduction.

3. **Reviewer** — 2) On page 20 (last lines) the authors make the point that the localization is an important effect and that its disregard in global models with their coarse vertical resolution and in weather models leads to "significant uncertainties". To my view, this is too cheap a statement. The statement may be ok if it had been written in conditional tense and without the "significant". Otherwise, it must be shown what the bad consequences of its negligence are on dehydration, radiation, water vapor transport into the tropical stratosphere, etc.

   **Authors** — We changed the statement following the reviewer's suggestion.

4. **Reviewer** — 1) Page 2, Line 21/22: As the wave phase is a purely mathematical object, I suppose that it can only affect ice crystals indirectly. An influence can only be exerted by material (physical) properties of the crystals environment, as T or RH. Does your statement imply that such properties are uniquely related to the wave phase?

***Authors*** — Yes, with our assumptions temperature and relative humidity anomalies are uniquely related to the wave phase.

5. ***Reviewer*** — 2) P. 3, L. 2: Is it possible at all that RH=const in a wavy environment? Perhaps in this special case you better speak of "solid particles that fall but that don't grow or sublimate" instead of ice.

   ***Authors*** — It might be possible within very thick, high ice crystal number clouds which would damp the relative humidity. But we agree with the reviewer's suggestion that it is better to talk of solid particles and changed the text accordingly.

6. ***Reviewer*** — 1) Page 1, Line 22/23: Isn't the wind identical to the movement of air parcels?

   ***Authors*** — Yes, but not to the motion of falling particles.

7. ***Reviewer*** — P. 2, L. 2: 190 K is not a range.

   ***Authors*** — Corrected

8. ***Reviewer*** — P. 2, L. 3: insert "of the" before "atmosphere".

   ***Authors*** — Corrected

9. ***Reviewer*** — 4) P. 2, L. 15: write "to and fro" instead of "to and from".

   ***Authors*** — Corrected

10. ***Reviewer*** — 5) P. 2, L. 20/21: "the falling particles fall in the same direction as the wave phase" implies that the phase falls. Better write "the falling particles fall in the direction of wave propagation".

    ***Authors*** — Corrected

11. ***Reviewer*** — 6) P. 2, L. 32: which system?

    ***Authors*** — The wave-ice crystal system. This has been precised.

12. ***Reviewer*** — 7) P. 5, L. 3,4: "green" should be "red".

    ***Authors*** — Corrected.

13. ***Reviewer*** — 8) P. 5, L. 26: Although the notion "perfect gas" exists (a further simplification of an ideal gas), the gas constant should be termed "gas constant" or "specific gas constant for water vapor". There is nothing in the calculation presented that needs the assumptions of a "perfect gas".

    ***Authors*** — Changed

14. ***Reviewer*** — 9) P. 7, L. 5: please write "crystal number concentrations".

    ***Authors*** — Changed

15. **Reviewer** — P. 7, L. 27: The "next section" is 2.2.3, not 2.2.1.

**Authors** — Corrected

16. **Reviewer** — 11) Eq. 18: I am puzzled by the terms RH i c (Z). Before RH i c was introduced as a constant. Why is it now a function of Z? Please mention also the meaning of the terms in the brackets (probably Clausius-Clapeyron and pressure change?).

**Authors** — We were keeping the $Z$ because that formula is valid for any profile of relative humidity. We now specify the different terms.

17. **Reviewer** — 12) P. 8, L. 17: As RH i c was never specified, is the fix point possible for the whole range of possible values? Is it tacitly to be understood that RH i c is close to or above ice saturation since there are ice crystals?

**Authors** — The necessary condition for the fixed point is that there exists regions where $RH_i = 100\%$ in the wave field. This has been clarified in the text (Eq. (22)).

18. **Reviewer** — P. 9, L. 3: Check sentence!

**Authors** — Rephrased

19. **Reviewer** — 14) P. 9, L. 13: Which of the amplitudes?

**Authors** — Temperature amplitude, this is now specified.

20. **Reviewer** — 15) Fig. 2: Is it possible to indicate the direction of the motion in phase space?

**Authors** — We have added arrows to indicate the direction.

21. **Reviewer** — 16) P. 11, L. 11-12: On first reading, it was not clear to me what exactly is the difference between the "cold" phase in the eastern Pacific and the "cooling" phase in the western Pacific. Only the later reference to figure 2 clarifies that. I suggest to refer earlier to the figure to illustrate the distinction.

**Authors** — Done.

22. **Reviewer** — 17) Figure 4 does not show green points, contrary to what the caption says.

**Authors** — Corrected

23. **Reviewer** — 18) P. 14, L. 17: blue is a color as well!

**Authors** — Corrected

24. **Reviewer** — 19) P. 16, last par: change "equality" to "equation".

**Authors** — Changed

25. **Reviewer** — 20) Figure 6: blue and black are hard to distinguish.

**Authors** — We dashed the blue line to avoid the confusion.

26. ***Reviewer*** — 21) P. 18, L. 15: The sentence is a bit strange. In clear sky there are no cirrus clouds. How can then their dehydration efficiency be constrained?

***Authors*** — We meant all sky dehydration efficiency, we have rephrased that sentence.

27. ***Reviewer*** — 22) P. 19, L. 6: "order" should be "power".

***Authors*** — We meant order moment. Corrected.

28. ***Reviewer*** — 23) P. 19, L 30.: "disagreement" between what?

***Authors*** — Models and observations. Now specified

**References**

Jensen, E. J., Diskin, G., Lawson, R. P., Lance, S., Bui, T. P., Hlavka, D., McGill, M., Pfister, L., Toon, O. B., and Gao, R.: Ice nucleation and dehydration in the Tropical Tropopause Layer, Proc. Nat. Acad. Sci., 110, 2041–2046, doi:10.1073/pnas.1217104110, 2013.

Jensen, E. J., Pfister, L., Jordan, D. E., Bui, T. V., Ueyama, R., Singh, H. B., Thornberry, T. D., Rollins, A. W., Gao, R.-S., Fahey, D. W., Rosenlof, K. H., Elkins, J. W., Diskin, G. S., DiGangi, J. P., Lawson, R. P., Woods, S., Atlas, E. L., Rodriguez, M. A. N., Wofsy, S. C., Pittman, J., Bardeen, C. G., Toon, O. B., Kindel, B. C., Newman, P. A., McGill, M. J., Hlavka, D. L., Lait, L. R., Schoeberl, M. R., Bergman, J. W., Selkirk, H. B., Alexander, M. J., Kim, J.-E., Lim, B. H., Stutz, J., and Pfeilsticker, K.: The NASA Airborne Tropical Tropopause Experiment: High-Altitude Aircraft Measurements in the Tropical Western Pacific, Bulletin of the American Meteorological Society, 98, 129–143, doi:10.1175/BAMS-D-14-00263.1, URL http://dx.doi.org/10.1175/BAMS-D-14-00263.1, 2017.

---

## Author Comment (AC2) · 9 Apr 2018

**Impact of gravity waves on the motion and distribution of atmospheric ice particles: reply to reviewer 2**

April 9, 2018

We would like to thank the reviewer for his/her very insightful comments and suggestions regarding our manuscript. Please find below our point-by-point reply.

1. **Reviewer** — Description and analysis of the simplified ODE system: Generally, it is a very meaningful approach to formulate a simple model for representing the important processes and to use this model for a rigorous analysis; this is also a very interesting and important result of this study.

   However, this part of the manuscript should be revised and partly rewritten, since it is very difficult to follow the line of arguments. This is mostly due to the very irritating notation, which is changed in the section several times. For instance, new coefficients as $\alpha_G$ are intro- duced but only partly used. Sometimes the text refers to "the first" or "the second" equation, but it is not really clear, which equations are meant. In fact, the restriction for the relative humidity to be equal to 100% does not follow from the "second" equation (19) but from the requirement for the equilibrium point, that the derivatives must be zero and thus the radius can only be constant, if the cloud is in thermodynamic equilibrium.

   **Authors** — We have rewritten this section to clarify the notations and the associated mathematical explanation, following the reviewer's suggestion (see also below).

2. **Reviewer** — Beside the confusing (but nevertheless correct) description of the model system and the linearization, there is a major problem for the correct analysis of the nonlinear system. The qualitative behaviour of the equilibrium point in the linearisation can only be transferred to the original nonlinear system, if the eigenvalues have non-zero real part (hyperbolic points). Thus, for the saddle point the argumentation is correct. For points with eigenvalues of zero real part (non-hyperbolic points), the quality of a centre point (in the linearization) cannot easily be transferred to the non-linear system (see, e.g., Verhulst, 1996 or Hirsch et al., 2013).

   I would suggest (also in terms of simplification of the notation) to rewrite the system using new variables $x = \Psi$, $y = r^2$ and constants a, b, c, d:

   $$\dot{x} = -c - dy(1)\dot{y} = -a sin x + b(2) \tag{1}$$

This abstract formulation helps to see the formal structure of the equations. In fact, it can be seen easily that the system is Hamiltonian with a Hamilton function as follows (transformation $q = x$, $p = y$):

$$H(p, q) = -cp - p^2 - bq - a \cos q \qquad (2)$$

Using the Hamilton function, the stability of the elliptic point as well as the existence of the periodic solutions can be determined easily. In addition, the Hamilton function might be used for the calculation of trajectories, since solutions are given by $H(p, q) = const.$, and maybe also for determining the domain of attraction around the elliptic point. This might be interesting in the sense, how many ice particles are really influenced by the mechanism, or better, how close the particles must be to the elliptic point to be affected.

***Authors*** — We are very grateful to the reviewer for the suggestions, especially for pointing out the Hamiltonian structure of the system. We agree that the analysis near the elliptic point was not rigorous (although supported by the numerical calculations); this is now corrected with the Hamiltonian formulation.

We have adopted the simplified notations suggested by the reviewer, except for the introduction of the new variables $x$ and $y$. We believe indeed that the physical meaning of the equations is better presented by expressing them with the physical quantities, and chose to prefer that point over a clearer presentation of the mathematical structure, which we put in a new appendix.

3. ***Reviewer*** — Neglecting water vapour depletion by ice crystals: For the formulation of the model equations (11) and also the simplified model (eq. 19), the depletion of water vapour by crystal growth is neglected. I can understand that for the analysis of the model this is a convenient simplification. However, it should be estimated how large the effect on the background fields as well as the solutions really is. This should be done analytically and/or using numerical simulations. Probably, the effect is really small and the assumption is meaningful but this must be shown.

***Authors*** — The reviewer raises an important limitation of our study. However, evaluating the impact of vapor depletion requires assumptions on the ice crystal number and/or the ice nucleation process, which would be arbitrary. Testing the sensitivity to different ice crystal number concentrations is beyond the scope of our study and would weaken the focus of the work. We hence prefer to avoid this discussion, and emphasize that the approximation is realistic for very low ice crystal number clouds. Furthermore, this idealized context notably enables us to highlight the role of the wave-localization effect, which on its own is able to maintain clouds at $RH_i \simeq 100\%$.

4. ***Reviewer*** — The whole study treats ice crystals, which are already there, i.e. the formation of ice crystals is not taken into account. However, in principle ice crystals are formed in the low temperature regime of TTL at high supersaturations (RHi $\sim$ 130 - 170%, depending on the formation mechanism). Thus, the assumption of ice crystals in a region at thermodynamic equilibrium seems to be quite strong. For me two different scenarios might be possible, if we start with ice nucleation: (a) If only a few ice crystal form, they are not able to deplete enough water vapour for reaching equilibrium and thus the described

mechanism does not work, until the ice crystals have grown to larger sizes and have fallen out into a region with relative humidity close to ice saturation. It is not clear if for large ice particles (radius close to 100 µm) the described mechanism will be efficient. Please, comment on this. (b) If many ice crystals are formed, they will deplete the water vapour without growing to larger sizes (because they are many) until the system reaches equilibrium. Then the described mechanism can play a role. In this scenario, please describe, how large the effect of wave-driven localization is in comparison to quenching of water vapour.

***Authors*** — We agree with the reviewer: we make the assumption that there are already some ice crystals in the region of thermodynamic equilibrium to start with. However, as mentioned above, including ice nucleation would require additional assumptions which we would like to avoid. Furthermore, it is not reasonable to include ice nucleation without also including the spectrum of high frequency gravity waves that influence its outcome (Spichtinger and Krämer, 2013). When those higher frequency waves are considered, the two scenarios proposed by the reviewer are not the only possibilities for crystals to get in regions of thermodynamic equilibrium of a lower frequency wave. Indeed, adding smaller scale waves might increase the relative humidity from $\sim 100\%$ to the heterogeneous or homogeneous threshold and thus trigger nucleation in those regions. For consistency, including nucleation would then also require a rigorous treatment of those waves. Although it might be possible to extend our framework to include the noise induced by high-frequency waves through deriving stochastic differential equations, this would require significant additional work and would also complicate the message of the paper. We thus prefer to leave those considerations for future studies and restrict ourselves to a discussion of the realism and possibility of generating ice crystals in the thermodynamic equilibirum region in the first place. This point is now discussed in Sect. 3.

5. ***Reviewer*** — Figure 1: Aspect ratio of the phenomenon In the example of figure 1, the vertical extension is of order O(3 km) whereas the horizontal extension is of order O(10 3 km); thus the aspect ratio is very small, please indicate this in the text and also in the figure caption.

   ***Authors*** — This is specified in the revised manuscript.

6. ***Reviewer*** — Page 4, lines 7-15 and following next page: It seems that the effect of wave-driven localization is mainly effective for waves with quite low frequencies (Kelvin waves). Please comment this in the text.

   ***Authors*** — Actually, the only requirement is on the vertical phase speed of the wave. However, the integrated effect on the displacement will be larger for low frequency waves (Kelvin waves or equatorial Inertio-Gravity waves), which is now specified in the text.

7. ***Reviewer*** — Constraining the value of deposition coefficient: Actually, Skrotzki et al. (2013) does give a recommendation for a value of the deposition coefficient, based on a collection laboratory experiments, model simulations and a synthesis of both, i.e. $0.2 \leq \alpha_d \leq 1$. Thus, the used value of $\alpha_d = 0.5$ is in the recommended range. Please reformulate the text accordingly.

   ***Authors*** — We modified the text accordingly.

8. **Reviewer** — Expression for the saturation mixing ratio: The correct (but still approximate) formula for the saturation mixing ratio is $qsat = \epsilon e_{sat}(T)/P$ with $\epsilon$ the ratio of molar masses of water and air, respectively.

    **Authors** — We actually are working with the volume mixing ratio rather than the mass mixing ratio. This is now specified in the text.

9. **Reviewer** — Figure 4 and text: In this figure the time evolution of the particles'position is shown. It would be nice to quantify how many particles from the initial distribution at 0.0 days really survive in a position close to the elliptic point. A similar statistics would be interesting for the simulations in figure 5 and figure B1 in the appendix.

    **Authors** — We agree with the reviewer, and now mention the statistics in the text.

10. **Reviewer** — Page 15, line 15 and equation (15): Slow down of ice crystal sedimentation It is stated here that the sedimentation is reduced significantly by wave advection. Can you quantify this statement, i.e. by which fraction is the sedimentation reduced for distinct conditions?

    **Authors** — This has been added.

11. **Reviewer** — Validity of several approximations For the formulation of the model equations some approximations are made without much information about the validity of the approximation, e.g. the assumption of spherical particles (Stokes'flow for sedimentation, eq. 17) or the linearisation of the saturation vapour pressure (eq. 18). Please indicate (at least in the appendix) the validity of these approximations quantitatively. On the other hand, the full growth factor for ice crystals is used, including kinetic and ventilation corrections and latent heat release. Since the model is used in a very small part of the phase space (radius 5 $\mu$m $\leq$ r $\leq$ 100 $\mu$m, very cold temperatures in the TTL) not all corrections are really meaningful or necessary. Thus, there is a kind of discrepancy between approximations on one hand and very accurate treatment of processes on the other hand. Please resolve this discrepancy in a meaningful way.

    **Authors** — In Sect. 2.2.1 the full equations are presented and they are the ones used for the numerical analysis in Sect. 3.1. The main microphysical approximation present at that point is that of spherical particles (necessary for tractability). Following the reviewer's comment, we now discuss the validity of this approximation relative to observations in Sect. 2.2.1, and quantify the associated uncertainty in Sect. 4.2.

    If it is true that further approximations (including the linearisation of the saturation vapor pressure) are made in Sect. 2.2.2., they are only introduced in order to derive the simplified system (toy model) for the theoretical analysis. They are only used in the context of this simplified system, and relaxed starting Sect. 3. Besides, for the toy model, we also approximate the growth factor to its expression without kinetic or ventilation corrections. Since approximations are introduced for the different microphysical factors at the same point, in our opinion, there is no real discrepancy between the treatment of the different processes.

**References**

Spichtinger, P. and Krämer, M.: Tropical tropopause ice clouds: a dynamic approach to the mystery of low crystal numbers, Atmos. Chem. Phys., 13, 9801–9818, doi:10.5194/acp-13-9801-2013, 2013.

---

## Referee Report (RR1)

Review of revised verion of
**Impact of gravity waves on the motion and distribution of atmospheric ice particles**
by Aurélian Podglajen et al.

**General comment:**
The manuscript has improved a lot, it is much clearer and thus is a meaningful contribution to ACP. My comments were addressed in aa appropriate way. I have only a few minor or technical points, which should be addressed before the manuscript can be accepted.

**Minor points:**

1. Section 2.1: Maybe you can indicate that the scenario of a constant ice crystal radius is equivalent to thermodynamic equilibrium.

2. Page 10: You mention the "center manifold" but you did not explain, what it is. Please include a description/definition of the term.

**Technical comments**

The entry Hirsch et al. (2013) is not correct in the references.

**References**

Hirsch, M., S. Smale, R. Devaney, 2013: Differential Equations, Dynamical Systems, and an Introduction to Chaos. Academic Press (Elsevier), Amsterdam.

---

## Author Response (AR2)

**Impact of gravity waves on the motion and distribution of atmospheric ice particles: reply to reviewer 1**

**May 31, 2018**

We would like to thank the reviewer for the comments and corrections. They have been taken into account in the revised version.

1. **Reviewer** — Although I am still sceptic about the overall relevance of the described localisation effect, I am content with the author's replies and additional explanations given in the new version. It is clear that a more complete simulation (including the formation of ice crystals and the exchange of water mass between the growing/shrinking crystals and the vapour phase) would involve new degrees of freedom and that the localisation effect could be masked by other processes. I am curious to see your work on this topic in the future.

For now, please make the following minor corrections: Page 6, line 18: delete one instance of "then"; P. 11, l. 14: it seems to me that the "<" should be inverted into ">" in the first eq.; P. 16, l. 7: "on" transport; P. 20, eq. 34 and the following text: The  $\rho$  in the square brackets should be "q" and in the text (line 1, P. 21) we have again  $F_ice$ , although that was replaced by  $F_{H20}$  in the equation. Please check and correct; P. 22, l. 29: delete one instance of "more".

Authors — We have corrected the mistakes reported by the referee.

2. **Reviewer** — In the figures, there is almost no contrast between the red and maroon colours. Please consider replacing maroon, e.g. by light blue.

Authors — We have replaced maroon by green.

**Impact of gravity waves on the motion and distribution of atmospheric ice particles: reply to reviewer 2**

**June 1, 2018**

We would like to thank the reviewer for the suggestions and corrections, which have been taken into account in the revised version.

1. **Reviewer** — The manuscript has improved a lot, it is much clearer and thus is a meaningful contribution to ACP. My comments were addressed in an appropriate way. I have only a few minor or technical points, which should be addressed before the manuscript can be accepted. Minor points: 1. Section 2.1: Maybe you can indicate that the scenario of a constant ice crystal radius is equivalent to thermodynamic equilibrium.

**Authors** — This was indicated in the first version of the manuscript, but the other reviewer found it confusing, because in a wave field it is hard to obtain thermodynamic equilibrium all along the crystals trajectory. That is why we prefer to refer to particles in general and not ice crystals in particular.

2. **Reviewer** — 2. Page 10: You mention the "center manifold" but you did not explain, what it is. Please include a description/definition of the term.

Authors — We removed that term which was unnecessary.

3. **Reviewer** — Technical comments The entry Hirsch et al. (2013) is not correct in the references

Authors — Now corrected

[revised manuscript text omitted]

tend to grow and leave that region to move into the positive temperature region, spending some time on the way in the cooling phase of the wave. Even outside of the center manifold, there are trajectories that remain near this if they are outside the region of permanent trapping near the elliptic point, some trajectories remain near that region a significant amount of time, so that the preferential location of crystals due to the *wave-driven localization* might manifest itself for a larger fraction of the crystal population than expected just from larger than that corresponding to the area enclosed by the red curves.

**2.2.3 Physical understanding**

5

10

The existence of the two fixed points can be easily understood physically. The elliptic point is located at  $RH_i = 100\%$  and  $\frac{\partial RH_i}{\partial z} < 0 \frac{\partial RH_i}{\partial z} > 0$ . Hence, if ice crystals fall below that fixed point, they fall into subsaturated air ( $RH_i < 100\%$ ) and sublimate, which reduces their mass and their fall velocity. They fall more slowly and are caught up again by the wave phase which is also descending. They may then be transported into supersaturated regions where they grow, increasing their weight

- and fall speed and moving them back into the equilibrium phase. The trajectory of an ice crystal in altitude-time space around the elliptic fixed point is sketched in Fig. 3, to illustrate the previous explanation. It can be noted that the ice crystal cycles around the elliptic point with a period of about 12 hours, consistent with Eq. (26) (see also Table 1). At the saddle fixed point, the reverse feedback is acting with subsaturated air above and supersaturated air below, so that the crystals move further away
- 15 from this equilibrium point.

In many aspects, the mechanism presented here is similar to the stabilization mechanism proposed by Luo et al. (2003) to explain the existence of Ultrathin Tropical Tropopause Clouds. Those authors considered ice crystals in a stationary vertical wind and relative humidity profile, and neglected the horizontal wind shear so that only vertical motions were examined. Then, a system of equations similar to (19) can be derived to describe the evolution of the crystal radius and position, where essentially

- 20 the altitude replaces  $\Psi$  and the vertical wind replaces the vertical wave phase speed. In that framework, for the same reason explained above for the elliptic point, ice crystals are stabilized in regions where  $RH_i = 100\%$ ,  $\frac{dRH_i}{dz} > 0$ ,  $\frac{\partial RH_i}{\partial z} > 0$  provided that the vertical wind is constant or decreasing with altitude. Taking a vertical profile, the location where the wave-driven localization occurs in our analysis (i.e., the elliptic point) is thus the same as the one where the stabilization effect of Luo et al. (2003) is expected ( $RH_i = 100\%$ ,  $\frac{dRH_i}{dz} > 0$ ,  $\frac{\partial RH_i}{\partial z} > 0$ ). However, since waves essentially dominate the variability of
- the vertical wind in the TTL, the setting of a quasi-monochromatic wave considered above is likely more realistic than the stationary vertical wind profile without horizontal wind shear used by Luo et al. (2003).